# Rapid fabrication of vascularized and innervated cell-laden bone models with biomimetic intrafibrillar collagen mineralization

Greeshma Thrivikraman [1], Avathamsa Athirasala[2], Ryan Gordon[3], Limin Zhang[3], Raymond Bergan [3], Douglas R. Keene[4], James M. Jones [5], Hua Xie[5], Zhiqiang Chen[6], Jinhui Tao[7], Brian Wingender [8], Laurie Gower[8], Jack L. Ferracane[1] & Luiz E. Bertassoni[1,2,5,9]

Bone tissue, by definition, is an organic–inorganic nanocomposite, where metabolically active cells are embedded within a matrix that is heavily calcified on the nanoscale. Currently, there are no strategies that replicate these definitive characteristics of bone tissue. Here we describe a biomimetic approach where a supersaturated calcium and phosphate medium is used in combination with a non-collagenous protein analog to direct the deposition of nanoscale apatite, both in the intra- and extrafibrillar spaces of collagen embedded with osteoprogenitor, vascular, and neural cells. This process enables engineering of bone models replicating the key hallmarks of the bone cellular and extracellular microenvironment, including its protein-guided biomineralization, nanostructure, vasculature, innervation, inherent osteoinductive properties (without exogenous supplements), and cell-homing effects on bone-targeting diseases, such as prostate cancer. Ultimately, this approach enables fabrication of bone-like tissue models with high levels of biomimicry that may have broad implications for disease modeling, drug discovery, and regenerative engineering.

[1] Division of Biomaterials and Biomechanics, Department of Restorative Dentistry, School of Dentistry, Oregon Health and Science University, Portland, OR 97201, USA. [2] Department of Biomedical Engineering, School of Medicine, Oregon Health and Science University, Portland, OR 97239, USA. [3] Division of Hematology/Oncology, Knight Cancer Institute, Oregon Health and Science University, Portland, OR 97239, USA. [4] Shriners Hospital for Children, Portland, OR 97239, USA. [5] Center for Regenerative Medicine, Oregon Health and Science University, Portland, OR 97239, USA. [6] Center for Electron Microscopy and Nanofabrication, Portland State University, Portland, OR 97201, USA. [7] Pacific Northwest National Laboratory, Richland, WA 99354, USA. [8] Department of Materials Science and Engineering, University of Florida, Gainesville, FL 32603, USA. [9] Cancer Early Detection Advanced Research (CEDAR), Knight Cancer Institute, Oregon Health and Science University, Portland, OR 97239, USA. Correspondence and requests for materials should be addressed to L.E.B. (email: bertasso@ohsu.edu)

The native bone extracellular matrix consists of an intricate structure constituted primarily of type I collagen fibrils co-assembled with non-collagenous proteins (NCPs) and apatite crystallites[1]. On the ultrastructural level, these crystals are arranged in the form of nanosized platelets that are hierarchically distributed both within (intrafibrillar) and between (extrafibrillar) collagen fibrils in the tissue matrix[2–5]. Given the outstanding load-bearing function of the bone, such an intricate hierarchical distribution of mineral has drawn significant attention in the materials engineering community and has been shown to be a key determinant to the long-range structure and function relationships of native bone[6,7]. In bone biomineralization, the deposition of apatite crystals inside collagen fibrils has been suggested to be synergistically orchestrated by matrix NCPs[8,9], the periodic arrangement of the tropocollagen molecules[2,10], fibril geometry[11], and water[12,13]. In-vitro model systems have led to the hypothesis that NCPs sequester mineral ions to form metastable, liquid-phase nanodroplets of amorphous calcium phosphate[14,15], which penetrate the interstices of collagen fibrils via capillary[16] and/or electrostatic interactions[17,18], later transforming into thermodynamically stable carbonated, calcium-deficient hydroxyapatite[19]. Efforts to controllably mimic the process of nanoscale bone biomineralization in vitro date back to decades ago and have had increasing levels of success[10,16,18,20–23]. These have included the use of poly(amino acids) and synthetic organic polyelectrolytes early on[24], and have recently explored the use of self-assembling peptide-amphiphiles[21,22] and anionic polymer acids to mimic the function of NCPs in templating hydroxyapatite growth within collagen fibrils in vitro[10,14,16,18,20,25]. These strategies, however, fail to mimic the cell-rich characteristic of mineralized bone and therefore have found limited use as biologically relevant model systems to study bone function, disease, cell-homing mechanisms, and response to drugs or repair.

Cell-based approaches to mimic human bone in the lab have relied primarily on the use of pre-calcified materials, such as brittle ceramics[26,27] or simulated body fluid-treated scaffolds that are post-seeded with osteoprogenitor cells[28]. Despite the outstanding relevance of these biomaterials for bone regeneration[29], these scaffolds utilize cells in two-dimensional monolayers seeded within relatively large pores and thus oversimplify the complexity of the three-dimensional (3D) microenvironment of the bone. Moreover, they are unable to accurately reproduce the gradual entrapment of osteoprogenitors in the bone matrix in the form of osteocytes, which represent over 90% of bone cells and regulate bone function in a paracrine manner, from inside-out[30]. Cell-laden polymeric hydrogels[31,32], which have been proposed as a superior alternative, approximate the 3D nature of the cell-rich bone matrix more closely. However, they too fail to replicate the complexity of bone's nanoscale calcification and mineral formation is typically restricted to small and dispersed nodules that appear after 14–21 days of culture in vitro[33]. Model systems that controllably replicate the true complexity of the bone tissue, including its nanoscale-calcified extracellular matrix microenvironment embedded with human cells, should allow for extensive experimental manipulation, tunability, and throughput, while also enabling improved analyses of cell response to essential cell–matrix and cell–mineral interactions that naturally occur in native bone. Such model systems currently do not exist.

Here we describe the encapsulation of undifferentiated human mesenchymal stem cells (hMSCs) in 3D microenvironments, cultured in supersaturated calcium and phosphate-rich cell media supplemented with a NCP analog, which directs the formation of nanoscale hydroxyapatite in the interstices of collagen fibrils. This process mimics the nanoscale structure, composition, and a set of important biological functions that are characteristic to the cell-rich calcified bone microenvironment. We show that the matrix nanoscale mineralization alone can stimulate the osteogenic differentiation of bone marrow-derived stem cells to levels that are comparable to those obtained with standard osteoinductive media. Moreover, this process leads to cell morphology and cell–matrix interactions that are consistent with the characteristics of pre-osteocytes embedded in mineralized bone. Lastly, we show that our proposed strategy enables the formation of pericyte-supported blood capillaries and integrated neuronal networks that are cemented within a bed of dense minerals, both of which address the long-standing challenges of engineering vascularized and innervated bone-like tissues in vitro. We further validate this model system by showing that the engineered tissue can stimulate homing of engrafted prostate cancer cells in vivo. In summary, we demonstrate that the proposed strategy allows for controlled engineering of a bone-like model system with a high level of nanoscale biomimicry and desirable biological functions, which we argue may have broad implications for drug discovery, regenerative medicine, and different aspects of bone research.

## Results

### Biomimetic intrafibrillar mineralization of cell-laden collagen.

In native bone tissue, the extracellular levels of Ca and P ions are supersaturated with respect to hydroxyapatite, so their precipitation is tightly controlled by anionic matrix proteins[9], which purportedly act as nucleation inhibitors. Our strategy works by supplementing soluble $Ca^{2+}$ and $PO_4^{3-}$ to the cell culture medium and then promoting a protein-induced intrafibrillar collagen mineralization process, using milk-extracted osteopontin (mOPN), an anionic protein, as a nucleation inhibitor. This process prevents spontaneous precipitation of calcium and phosphate in the medium, while modulating the non-classical (i.e., amorphous precursor) nanoscale mineralization[34] within the collagen fibrils throughout the cell-laden matrix in a rapid fashion. We first screened cell medium containing varying molar concentrations of $Ca^{2+}$ (1.125–18 mM) and $PO_4^{3-}$ (0.525–8.4 mM) in combination with different concentrations of mOPN (1–1000 μg/mL) to test for cell compatibility (Supplementary Fig. 1). High concentrations of $Ca^{2+}$ and $PO_4^{3-}$ had a significant cytotoxic effect, which can be attributed to the osmotic imbalance between the cell cytoplasm and the medium. When ions were complexed with mOPN, however, such cytotoxic effects were significantly reduced. We also determined the rate of mineral formation in collagen hydrogels by comparing values within a lower range of $Ca^{2+}$ and $PO_4^{3-}$ concentrations that were not cytotoxic (Supplementary Fig. 2). Within such a range, cell medium supplemented with 4.5 mM $Ca^{2+}$ and 2.1 mM $PO_4^{3-}$, stabilized with 100 μg/mL mOPN, which are in the higher range of levels of $Ca^{2+}$ and $PO_4^{3-}$ typically reported for extracellular fluids in the body[35], were chosen for the remainder of the experiments due to the greater efficiency of matrix mineralization and lack of cytotoxicity toward hMSCs. Next, cell-laden mineralized tissue constructs were generated by encapsulating hMSCs in a fibrillar collagen hydrogel (1.5 mg/mL), which were subsequently exposed to the supersaturated calcium and phosphate solution stabilized with mOPN at the aforementioned concentrations for 3 days, so as to induce both intra- and extrafibrillar nanoscale mineralization in the matrix surrounding the cells.

We first determined the ability of our 3-day mineralization protocol to mimic the nanoscale structure and mineral composition of native bone. Scanning electron microscopy (SEM) analyses of non-mineralized controls (Fig. 1a) showed a visible distinction from mineralized samples (Fig. 1b), pointing to the presence of homogeneously distributed extrafibrillar calcium and phosphate deposits (Supplementary Fig. 3). Energy-dispersive X-ray

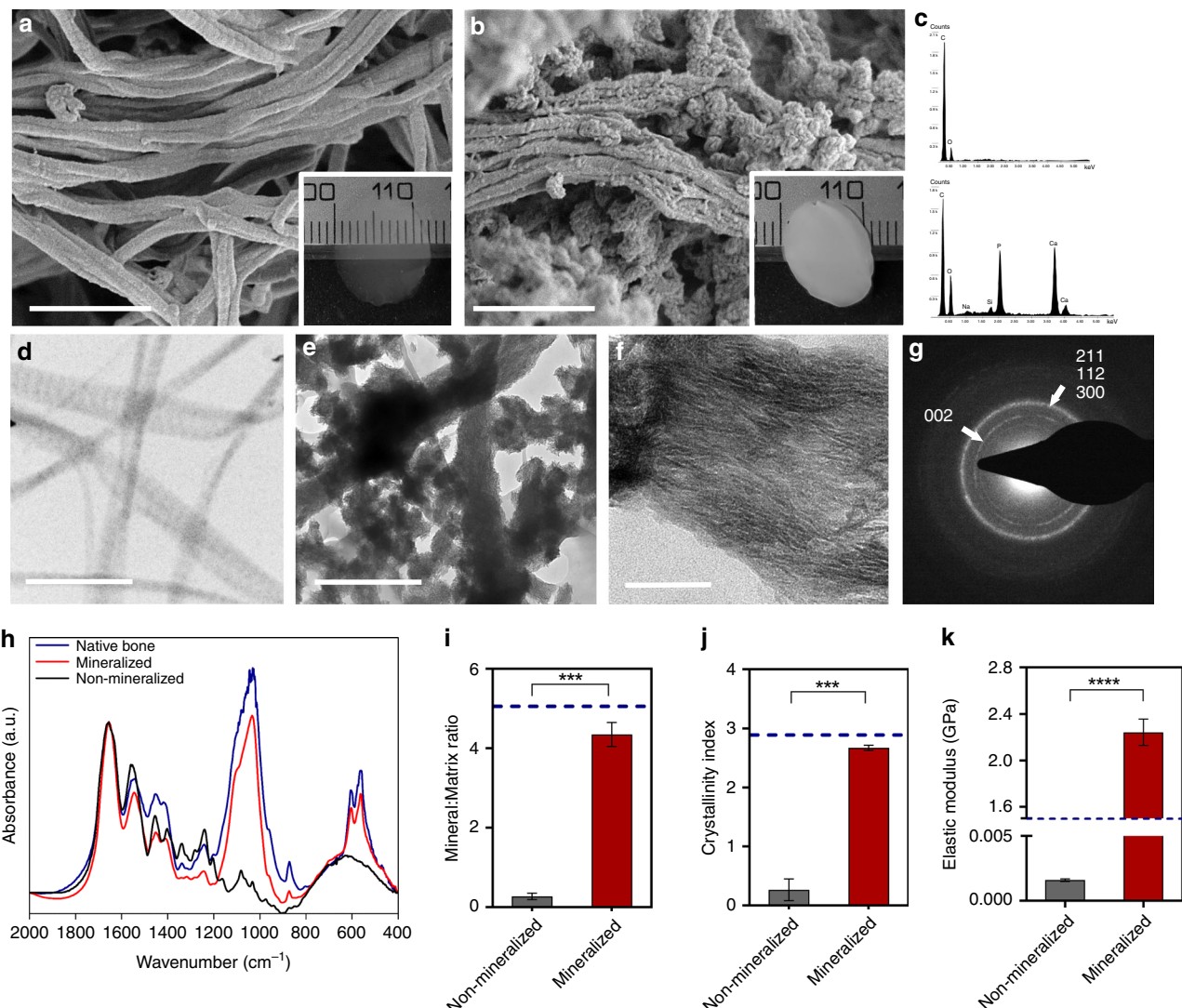

**Fig. 1** Nanostructure, composition, and characterization of mineralized collagen. SEM images of (**a**) non-mineralized and (**b**) mineralized collagen showing the formation of fibril bundles. Extrafibrillar mineral is apparent in **b**. ($N = 4$). Scale bar: 400 nm. Insets: Collagen hydrogels prior to (**a**) and immediately after (**b**) a 3-day mineralization period. The mineral formation resulted in a white opaque appearance. **c** EDX spectra of mineralized samples confirmed the presence of Ca and P in mineralized specimens (bottom), and lack thereof in non-mineralized controls (top). ($N = 4$). TEM images of (**d**) non-mineralized collagen showed the typical D-banding pattern of collagen, while the banding becomes obscured in the (**e**) mineralized collagen, which appears to have both intra and extrafibrillar mineral crystallites (Supplementary Movie 1). Scale bar: 500 nm. ($N = 4$). **f** The zoomed view of a single mineralized fibril suggests the preferential orientation of intrafibrillar mineral apatite crystallites (dark streaks) in the (001) position, parallel to the c-axis of the fibril. Scale bar: 50 nm. **g** Selective area electron diffraction (SAED) of a mineralized collagen shows the typical broad arcs for the (002) plane and overlapping arcs for the (112), (211), and (300) planes that are consistent with the known hexagonal crystalline structure of hydroxyapatite. **h** FTIR spectra and respective (**i**) mineral:matrix ratio (****$p < 0.0001$, Student's $t$-test) and (**j**) crystallinity index (***$p < 0.001$, Student's $t$-test) of mineralized and non-mineralized collagen constructs, dotted lines are reference values of native bone. Data are represented as mean ± SD ($N = 6$). **k** AFM nanoindentation modulus of non-mineralized and mineralized hydrated collagen fibrils. Mineralization resulted in over a 1000-fold increase in stiffness for individual fibrils (****$p > 0.0001$, Student's $t$-test) ($N = 3$). Dotted line is a reference value of native mineralized collagen[51]. Source data are provided as a Source Data file

spectroscopy (EDX) profiling of mineralized samples revealed Ca and P as the major constituents of the matrix (Fig. 1c, Supplementary Fig. 3). Furthermore, transmission electron microscopic (TEM) imaging of mineralized samples confirmed the additional intrafibrillar localization of apatite crystallites, with nanosized platelets visibly oriented in the (001) plane parallel to the long axis of the fibrils (Fig. 1e, f and Supplementary Figs. 4–6) hiding the typical banding pattern of non-mineralized collagen type 1 (Fig. 1d). Selective area electron diffraction (SAED) analyses of mineralized fibrils also revealed the typical broad arcs for the (002) plane and overlapping arcs for the (112), (211), and (300) planes, all of which are consistent with the known

hexagonal crystal form of hydroxyapatite in the native bone and in osteoblast-secreted apatite crystals (Fig. 1g and Supplementary Fig. 4)[3,36]. For comparison, we then generated 3D tomographic images of mineralized collagen fibrils by collecting a series of images as the TEM stage was tilted from +70 to −70° (Supplementary Movie 1). The ultrastructural features of the mineralized constructs revealed mineral accumulation in both intra- and extrafibrillar compartments, with a substantial amount of the mineral being situated in the extrafibrillar space (Fig. 1e and Supplementary Figs. 4–6), akin to native bone (Supplementary Fig. 7). We further characterized the chemical composition of our mineralized samples in comparison with that of native bone

using Fourier Transform infrared spectroscopy (FTIR). The peak intensities and positions of apatitic phosphate (1030, 600, and 560 cm$^{-1}$), carbonate (874 cm$^{-1}$) and amide I, II, and III (1649, 1553, and 1248 cm$^{-1}$) bands for the mineralized matrix were found to be similar to that of bone tissue (Fig. 1h). Similarly, although the absolute levels of mineral and collagen are inferior to those in native bone, given the low density of the collagen hydrogels in comparison with that of the native tissue, the relative mineral-to-matrix ratio (Fig. 1i) and crystallinity index (Fig. 1j) of the mineralized scaffolds approximated those of human bone. To determine the ability of the apatite mineral to bind to and mechanically reinforce the fibrils, we performed atomic force microscopy (AFM)-based nanoindentation on individual collagen fibrils in solution and ambient air (Supplementary Fig. 8), either before or after mineralization. The hydrated non-mineralized collagen had an elastic modulus of 0.0016 ± 0.0003 GPa, whereas the elastic modulus of mineralized fibrils was 2.21 ± 0.046 GPa (Fig. 1k). Despite the significant increase in nanoindentation modulus of individual fibrils, the overall stiffness of the hydrogel constructs was still markedly lower than that of the bone, which is in the order of 20 GPa on a macroscale (Supplementary Fig. 8). Moreover, the fibrillar organization, diameter, and D-banding periodicity of the collagen matrix were also reminiscent of that in native bone (Supplementary Fig. 9). Overall, both the mineral composition and nanostructural organization approximate that of a loosely packed, woven bone tissue.

Next, we evaluated whether the nanoscale mineralization of cell-laden hydrogels could lead to reduced cell viability due to increased osmotic damage or physical impairment of nutrient delivery to the cells in the matrix. Approximately 90% of cells embedded in the mineralized hydrogels remained viable after at least 7 days of culture in vitro, which was similar to both non-mineralized collagen and collagen treated with osteoinductive medium (OIM) as controls (Fig. 2a–d). We then evaluated whether the mineralization process could influence the generation of intracellular stress and the proliferation of cells in the cell-laden tissue constructs. During the 3 days of exposure to the guided mineralization process, there was no significant increase in the levels of reactive oxygen species (ROS) on day 1 and a comparable increase was found on day 3 where both mineralized and OIM-treated samples generated more ROS than non-mineralized controls. After 7 days, both treatments had comparable levels of ROS as non-mineralized controls and all groups were significantly lower than $H_2O_2$, which was used as a positive control for stress generation on the cells (Fig. 2e). Cell proliferation between non-mineralized, mineralized, and OIM-treated groups were comparable from day 1 to day 3, and after 7 and 14 days cell proliferation was still present but significantly higher in mineralized samples (Fig. 2f). On day 21, proliferation decreased in all three groups, potentially due to lineage commitment of cells to an osteogenic phenotype or growth impairment due to cell–cell contact. These findings confirm that the cell-laden hydrogel mineralization process is non-cytotoxic, and that neither cell metabolism nor viability are significantly affected, despite the dense calcification of the matrix surrounding the cells.

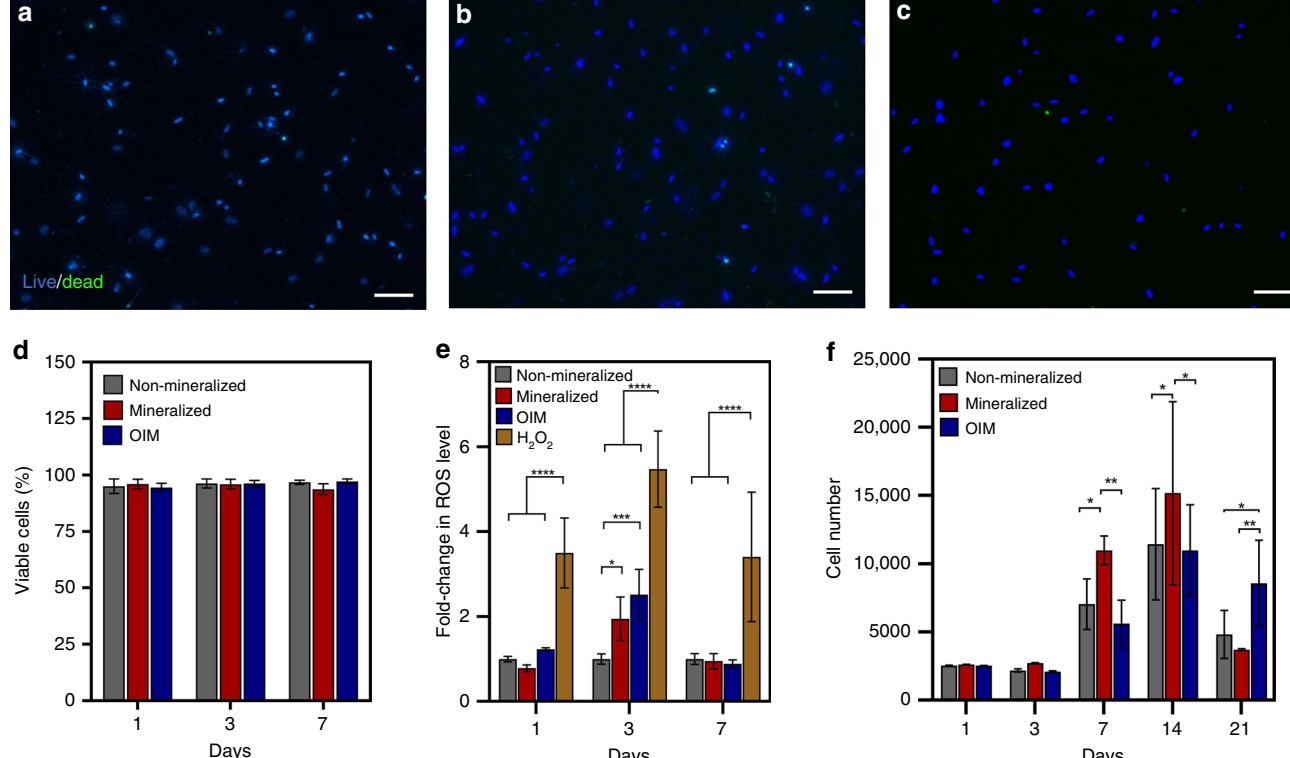

**Fig. 2** Cytocompatibility of the biomineralization process. Representative fluorescent micrographs of live (blue) and dead (green) stained hMSCs embedded in (**a**) non-mineralized, (**b**) mineralized, and (**c**) OIM-treated hydrogels after 7 days. Scale bar: 100 μm. **d** All groups showed more than 90% cell viability on days 1, 3, and 7, irrespective of the treatment method. Data are represented as mean ± SD ($N = 6$). **e** Mineralization of cell-laden hydrogels resulted in nonsignificant generation of reactive oxygen species (ROS) compared with $H_2O_2$ (*$p < 0.05$ and ****$p < 0.0001$, ANOVA/Tukey) after 7 days ($N = 6$). **f** Proliferation rate of hMSCs encapsulated in mineralized hydrogels was comparable to that of cells embedded in non-mineralized and OIM-treated hydrogels up to day 3 and was significantly higher than both groups on days 7 and 14 (*$p < 0.05$, **$p < 0.01$, ANOVA/Tukey). After 21 days, a sharp decrease in cell number was recorded in all groups, with a significant reduction observed in non-mineralized (*$p < 0.05$ ANOVA/Tukey) and mineralized (**$p < 0.01$ ANOVA/Tukey) hydrogels in comparison with OIM-treated samples ($N = 6$). Source data are provided as a Source Data file

**Biomimetic mineralization stimulates osteogenic differentiation.** It is well accepted that cells respond strongly to the structure and mechanics of the matrix in which they are embedded[37]. Matrix stiffness, especially at the single fibril level, is linked to important mechanisms of mechanotransduction-mediated cell differentiation[38]. Therefore, we expected that the expression of key osteogenic markers would be elevated soon after mineralization was completed, even without the aid of any osteogenic supplements. To assess that, we surveyed the transcription levels of major genes and proteins associated with either osteoblastic/pre-osteocytic differentiation, or bone metabolism and remodeling. These expression levels were compared against cells cultured either in non-mineralized controls or in collagen cultured in the presence of OIM containing ascorbic acid, dexamethasone, and β-glycerol phosphate, which are known to stimulate osteogenic differentiation of hMSCs. Cells in the mineralized matrix exhibited either significantly higher or comparable gene expression profiles to those obtained using OIM (Fig. 3a and Supplementary Fig. 10), with the exception of runt-related transcription factor 2 (*RUNX2*). The expression levels of osteocalcin (*OCN*), a late-stage marker for osteoblastic differentiation, had significantly higher expression in mineralized samples than the positive control after 21 days, wherein nearly a twofold increase was observed[39,40].

Similar to the expression of early-stage differentiation markers, matrix mineralization alone induced an early onset in the expression of *DMP1*, a marker that has been shown to be predominantly expressed in osteocytes but not in osteoblasts[41]. This was nearly four times higher than cells cultured in OIM after 7 days and still significantly higher on day 14. The expression of *PDPN*, a mucin-type protein highly expressed in osteocytes, was also comparable between mineralized samples and positive control at all time points. These results support the conjecture that mineralization of the surrounding matrix is a key determinant for osteoblast-to-osteocyte transition, both in vitro[42] and in vivo[43].

As both osteoblasts and osteocytes synthesize proteins that participate in bone homeostasis in a paracrine manner[30,44,45], we then analyzed the expression of a set of key proteins involved in bone metabolism (BMP-2, -6, and -7, DKK-1 and TGF-β), and remodeling (MMP-3, OPG, and RANKL, RANKL/OPG) (Fig. 3b). Of the three osteoinductive BMPs tested, the expression of BMP-2 was significantly higher in both the mineralized hydrogels and OIM when compared with non-mineralized controls. On the other hand, a significant increase in the secretion of BMP-6 was detected only in the mineralized samples. Conversely, an increased expression of DKK-1, an osteogenic inhibitor, was found in cells treated with OIM, but not in the mineralized constructs. With respect to the factors that are involved in bone remodeling, a fivefold drop in expression of osteoprotegerin (OPG) was noted in the mineralized constructs compared with both other groups (Fig. 3b). The expression of RANKL, which is one of the crucial cytokines for osteoclastogenesis[46], was significantly higher in both mineralized and positive control samples when compared with non-mineralized control. Furthermore, given that the balance between RANKL and OPG is a key determinant to bone resorption[47], we calculated the ratio of RANKL to OPG in our samples and found a marked increase for the mineralized group, which points toward the ability of cells embedded in the mineralized matrix to secrete soluble factors that may help orchestrate remodeling in a paracrine fashion, as it happens in bone.

To further validate the protein expression of osteoblastic and pre-osteocytic markers, we labeled cells for OCN, PDPN, and DMP1 via immunostaining after 7, 14, and 21 days (Fig. 3c–k and Supplementary Figs. 12–14). Consistent with our gene expression analysis, all three markers were poorly expressed in non-mineralized hydrogels, but showed a marked increase in expression after 7, 14, and 21 days for both the mineralized hydrogels and the positive control, although at different levels. These results correlated with the amounts of calcium, as seen by alizarin red staining on day 7, present in the matrix (Fig. 3l–n). Collectively, the gene and protein expression patterns suggest that matrix mineralization alone can stimulate the expression of both osteogenic and pre-osteocytic markers to levels that are at least comparable to gold standard osteoinductive supplements (OIM).

In native bone, osteoblasts go through a series of morphological changes as the surrounding matrix transforms from soft osteoid to more heavily mineralized bone[30]. Consistent with that notion, fluorescence images obtained from cells cultured for at least 7 days in the mineralized microenvironment showed cells extending narrow actin-rich dendritic-like processes that appeared to radiate through the matrix toward neighboring cells, and in some cases, with narrow extensions stemming from them (Fig. 3p). These morphological features resemble the characteristic morphology of osteocytes, where cell processes precede and guide dendrite growth and orientation[48]. In non-mineralized (Fig. 3o) or OIM-treated controls (Fig. 3q), on the other hand, cells lacked any marked increase in the projection of cell dendrites (Supplementary Fig. 15). Furthermore, images of the collagen matrix obtained in confocal reflectance mode revealed visible collagen fiber organization in the non-mineralized (Fig. 3r) and OIM-treated controls (Fig. 3t), whereas well-defined fluorescentless features that appeared devoid of collagen and resembled the appearance of bone lacunae were noted in the mineralized constructs (Fig. 3s).

To better understand the structural crosstalk between the encapsulated cells and the matrix after the mineralization process, we used a combination of serial-section backscatter electron (BSE) imaging and 3D digital reconstruction to simultaneously elucidate the ultrastructure of mineralized fibrils along with the microscale architecture of the embedded cells. As a first step, we acquired images from cells embedded in non-mineralized (Fig. 4a) and mineralized collagen (Fig. 4b), and in OIM-treated hydrogels (Fig. 4c). In the backscattered SEM images, the dark contrast is generated by the backscattered electrons emitted by minerals or denser features (i.e. cells and organelles), whereas the lower contrast areas are non-mineralized collagen or void spaces. Mineralized samples (Fig. 4b) had markedly dark contrasting fibrils throughout the matrix, particularly in the regions bordering the cells. These highly-mineralized regions are reminiscent of the *lamina limitans* that is seen around osteocytes in osteonal bone[49]. Next, we generated a series of images of mineralized samples at Z-intervals of 60 nm, with the intent of recreating a 3D digital image of the mineralized samples as a function of the contrast generated by the BSEs. We then used a set of 190 slices to digitally segment the cells, the mineral-free collagen, and the mineralized fibrils independently, based upon their respective electron-density contrast difference (Fig. 4d). A video of the orthogonal XYZ planes of these digital reconstructions is shown in Supplementary Movie 2. When viewed in 3D, cells are seen with a well-spread morphology, lying within a bed of densely packed mineralized fibrils (Fig. 4e, f). Of note, these fibrils are mineralized with similar levels of crystallinity as those observed in native bone and in osteoblast-secreted minerals (Supplementary Figs. 4–6). Cells interacted closely with the mineral and extended dendrite-like projections that are characteristic of an osteocyte-like phenotype (Fig. 4g). These long cell processes are consistent with the ones visualized in actin-stained cells, shown in Fig. 3p. Interestingly, regions adjacent to the embedded cells appeared more densely compacted with mineral (Fig. 4f). This indicates that even though ~50% of the organic

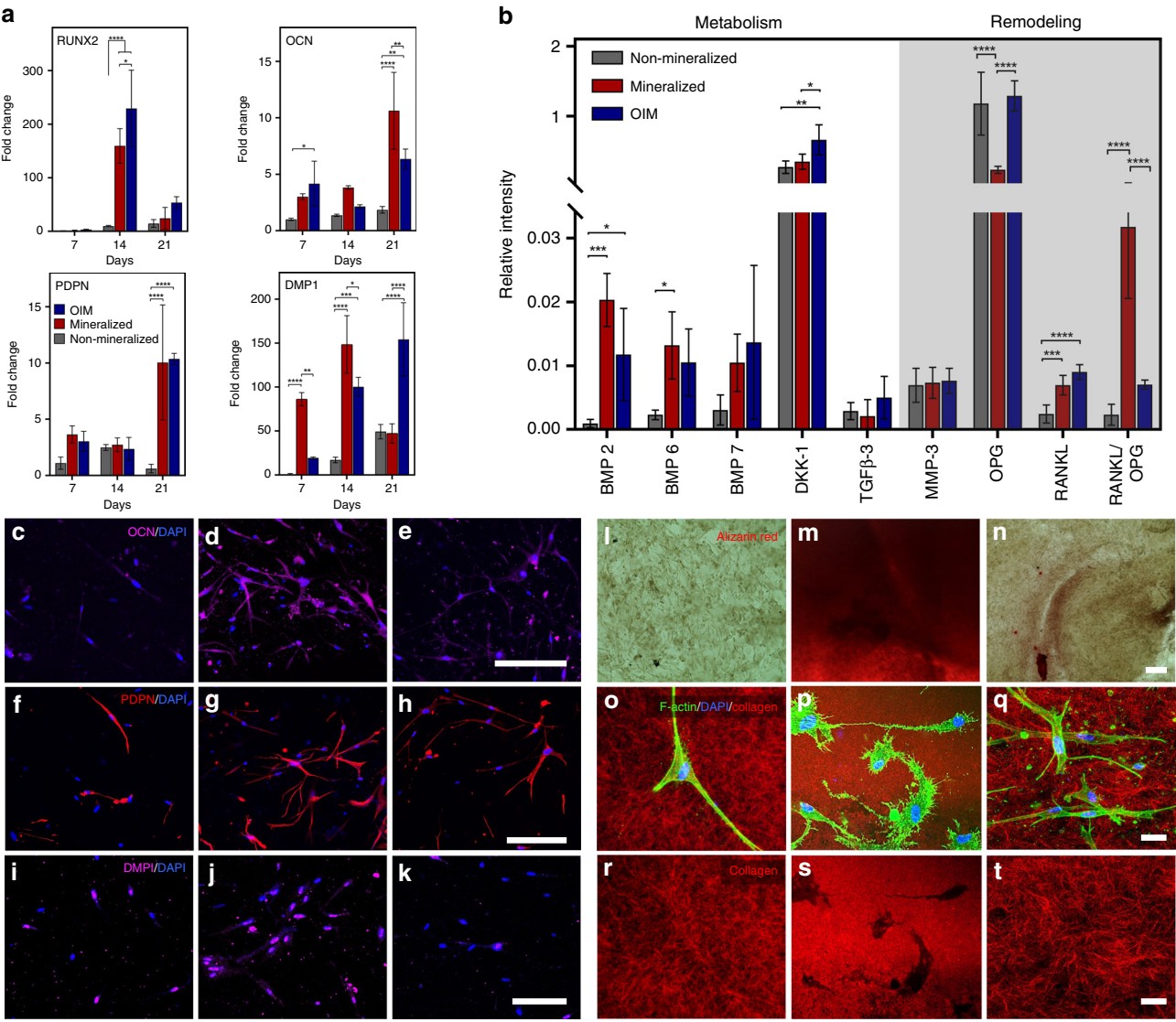

**Fig. 3** Osteogenic lineage commitment, gene, and protein expression in mineralized cell-laden collagen. **a** Gene expression of osteocalcin (*OCN*) was significantly higher (**$p < 0.01$) in mineralized collagen than in OIM treated, both of which were much higher than non-mineralized samples (****$p < 0.0001$), after 21 days. Expression of osteocyte-related genes (*DMP1* and *PDPN*) was also comparable to OIM after 21 days of culture in mineralized collagen and significantly higher at earlier time points (*DMP1*, **$p < 0.01$ after 7 days and *$p < 0.05$ after 14 days). Data are represented as mean ± SD ($N = 3$). **b** A significant increase in the expression of BMP-2 and BMP-6 for cells in both mineralized collagen and in OIM relative to non-mineralized controls is consistent with enhanced bone-specific metabolic activity. A marked increase in the ratio of RANKL/OPG in mineralized samples, however, suggests the stronger potency for cell-mediated bone remodeling via a paracrine signaling in cell-laden mineralized constructs than in the other groups. ($N = 4$ for BMPs and $N = 5$ for others). At a cell-surface level, the expression of OCN and PDPN on day 14 was very low in (**c**, **f**) non-mineralized controls and significantly higher for both (**d**, **g**) mineralized and (**e**, **h**) OIM-treated samples (**$p < 0.01$, ***$p < 0.001$) ($N = 3$). Similarly, surface expression of DMP1 was comparable for (**k**) OIM-treated cells (****$p < 0.0001$) and (**j**) mineralized collagen (*$p < 0.05$), both of which were significantly higher than (**i**) non-mineralized controls ($N = 4$). Scale bar: 200 μm. Alizarin red staining to assay mineralization shows homogeneous and intense red staining even after 7 days in (**m**) mineralized constructs, whereas staining was more diffuse in (**n**) OIM-treated samples and almost non-existent in (**l**) non-mineralized controls. Scale bar: 100 μm. Reflectance confocal microscopy images of F-Actin/DAPI-stained hMSCs in (**o**) non-mineralized, (**p**) mineralized, and (**q**) OIM treated cell-laden collagen, illustrate the dendritic-like extensions of cells after matrix mineralization, reminiscent of osteocyte-like morphology. The above images, with cells digitally removed (**r**–**t**), show formation of well-defined lacunae-like regions in the locations where cells resided in mineralized matrix (**s**). Scale bar: 30 μm. Quantification of dendrite-like projections is shown in Supplementary Fig. 15. All comparisons used ANOVA/Tukey. Source data are provided as a Source Data file

matrix was mineralized (Supplementary Fig. 16), cells were still able to move within the surrounding matrix (Supplementary Fig. 17 and Supplementary Movies 3, 4), secrete soluble proteins, as well as process intracellular and extracellular calcium (Supplementary Fig. 18), all of which are indicative of active new tissue formation. Overall, our results suggest that, when embedded in a microenvironment that replicates the three-dimensionality, composition and nanoscale structure of the mineralized bone niche, hMSCs expressed a multitude of morphological characteristics that are consistent with maturing bone cells, all in the absence of osteoinductive factors and driven primarily by matrix mineralization.

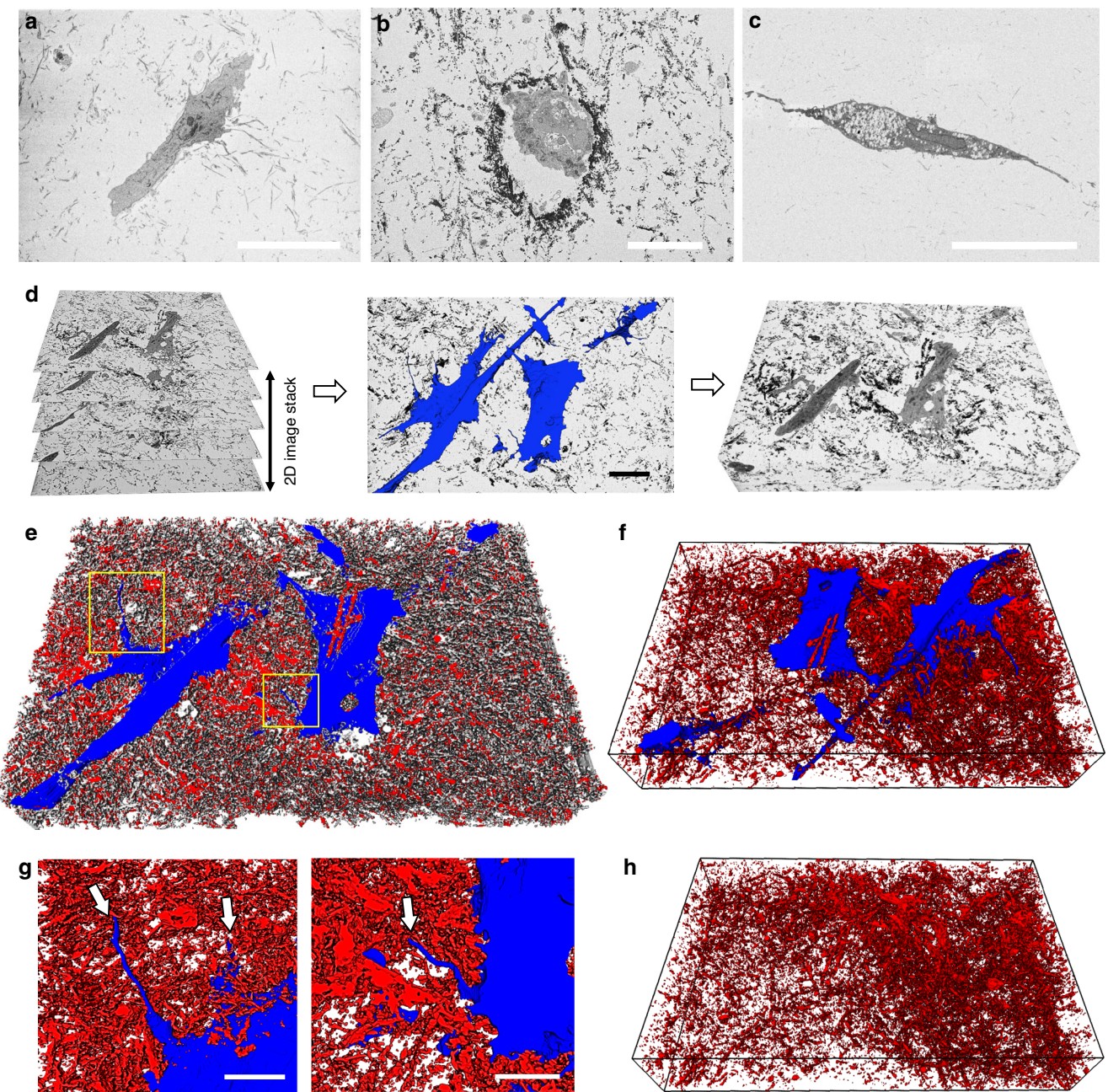

**Fig. 4** 3D volumetric reconstruction of BSE micrographs obtained via serial block-face SEM. **a** Matrix surrounding cells in non-mineralized collagen had little backscattered contrast, suggestive of lack of mineralization. **b** In mineralized hydrogels, the matrix was visibly darker due to the backscattered electron contrast of mineralized fibrils, especially in the matrix immediately surrounding the cells. **c** Collagen in OIM-treated samples also lacked significant backscattered electron signal. **d** Illustration of the serial stacking of 190 60 nm-thin sections, the segmentation of cells (blue) from the surrounding mineralized matrix (middle panel, scale bar: 20 μm), and visualization of block 3D image (right panel). Arrows in **d** show narrow dendrite-like cell processes. **e** 3D-rendered image of mineralized samples showing cells (blue) embedded in mineral (red), with the underlying collagen (gray). **f** Exclusion of collagen via digital processing in these mineralized samples illustrates the density of mineralized collagen and cells spread within a bed of mineralized matrix. Narrow cell processes (arrows) shown in higher magnification in **g** appear to extend between mineralized fibrils (Supplementary Movie 2) (scale bar: 10 μm). **h** Digital removal of cell bodies from within the mineralized matrix illustrates density of mineral surrounding the cell structures. The total length of the *x*-axis in all 3D reconstructions (**d**, **e**, **f**, and **h**) is 62 μm

**Microvascular/neuronal networks unaltered by mineralization.** In view of the versatility of our biomimetic strategy to trigger mineralization in a controlled manner at different time points (Supplementary Fig. 19), we co-cultured hMSCs with either neuroblastoma (SH-SY5Y) cells or with human umbilical vein endothelial cells (HUVECs) embedded in collagen hydrogels and allowed neuronal (Fig. 5) and vascular (Fig. 6) networks to form. We then subjected the innervated or vascularized constructs to

our process of nanoscale mineralization (Figs. 5a, 6a). Of note, during bone formation, vasculature and innervation form prior to the onset of calcification and the presence of extracellular Ca and P does not impair neither vasculogenesis nor neurogenesis. Thus, we set out to determine whether similar outcomes would be present in our system. Formation of 18-day-old neuronal networks interconnected by actin-rich neurites was indicated by the expression of neuron-specific enolase and neurofilament light

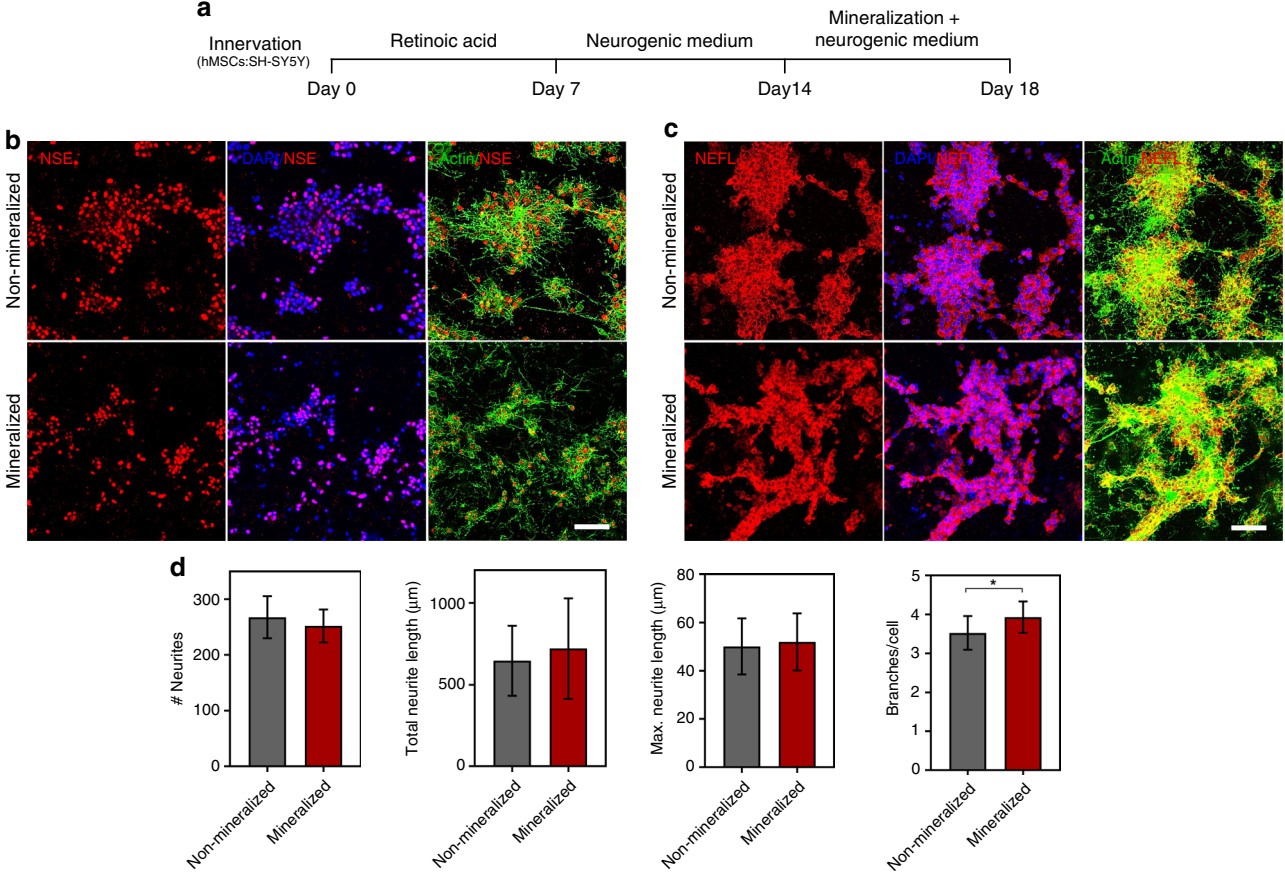

**Fig. 5** Innervation of mineralized cell-laden collagen. **a** Timeline for the generation of innervated and mineralized collagen constructs. Human SH-SY5Y neuroblastoma cells were co-encapsulated with hMSCs (4:1) in non-mineralized and mineralized hydrogels, after 14 days of differentiation. For the first 7 days, the cells were treated in DMEM/F-12 basal medium supplemented with 1% FBS and 10 μM all-*trans* retinoic acid (RA), followed by an additional 7 days of culture in Neurobasal-A medium containing 1% (v/v) L-glutamine, 1× B-27 supplement, 50 ng/mL human BDNF, and 10 μM RA. Subsequently, matrix mineralization was triggered by switching the neurobasal medium for the mineralizing medium for another 3 days. The fully differentiated neuronal cells within the mineralized constructs were confirmed by immunostaining with antibodies against Neuron-specific enolase (NSE) and Neurofilament light (NEFL). Cell nuclei were counterstained with DAPI (blue) and cytoskeletal actin was stained with Alexa Fluor 488 phalloidin (green). **b, c** Representative immunofluorescence images showing the expression of NSE and NEFL in non-mineralized and mineralized constructs. Both the neuronal differentiation markers had similar expression levels in mineralized and non-mineralized groups. Scale bar: 50 μm. **d** Imaris Filament Tracer module was used to quantify the morphological parameters of the differentiated SHSY-5Y cells in non-mineralized vs. mineralized constructs. Quantification of the number and length of neurites indicate no significant difference between non-mineralized and mineralized groups, whereas the number of branches and branch points was higher in the mineralized groups. Data are represented as Mean ± SD, *$p < 0.05$, Student's *t*-test ($N = 4$). Source data are provided as a Source Data file

(NEFL), which were visible in both mineralized and non-mineralized constructs at comparable levels (Fig. 5b, c). Quantification of number of neurites, total neurite length, and maximum neurite length were also statistically comparable between mineralized and non-mineralized hydrogels. The number of branches and branch points per cell, on the other hand, were significantly higher for cells embedded in the mineralized constructs (Fig. 5d). This suggests that the intrafibrillar collagen mineralization process does not hamper the formation of 3D neuronal networks in vitro.

Next, we hypothesized that vascular networks could be engineered in the core of the collagen scaffolds via endothelial cell morphogenesis, and that such constructs could then be mineralized to form vascularized bone-like tissue constructs that mimicked both the native bone vasculature and nanoscale mineralization. HUVECs and surrounding hMSCs formed vascular tubes within 3 days after cell encapsulation in the collagen hydrogels (Fig. 6b). The introduction of the Ca-, P-, and mOPN-supplemented medium induced homogenous mineral deposition throughout the matrix in 3 days, and HUVECs

encapsulated within mineralized hydrogels maintained the interconnected networks (Fig. 6b and Supplementary Fig. 20) and remained strongly positive for the endothelial cell junctional marker, CD31 (Fig. 6b). In addition, hMSCs that co-aligned with the endothelial tubes had a marked expression of α-smooth muscle actin (αSMA), which is a marker for differentiation of hMSCs into a pericyte-like phenotype (Fig. 6b and Supplementary Fig. 20). Of note, differentiation of hMSCs into αSMA-expressing cells in the mineralized constructs appeared to be restricted to cells in immediate contact with endothelial capillaries, whereas hMSCs that appeared to be away from the forming vessels maintained their capacity to differentiate into osteoblasts, as indicated by the expression of RUNX2 adjacent to green fluorescent protein (GFP)-expressing HUVECs (Fig. 6b). This was consistent with the presence of DMP1[+] cells near vascular capillaries in samples collected 7 days post implantation in vivo (Supplementary Fig. 21).

In order to determine the stability of our engineered vascular networks in vivo, we implanted the vascularized and mineralized constructs in the subcutaneous pockets of immunodeficient SCID

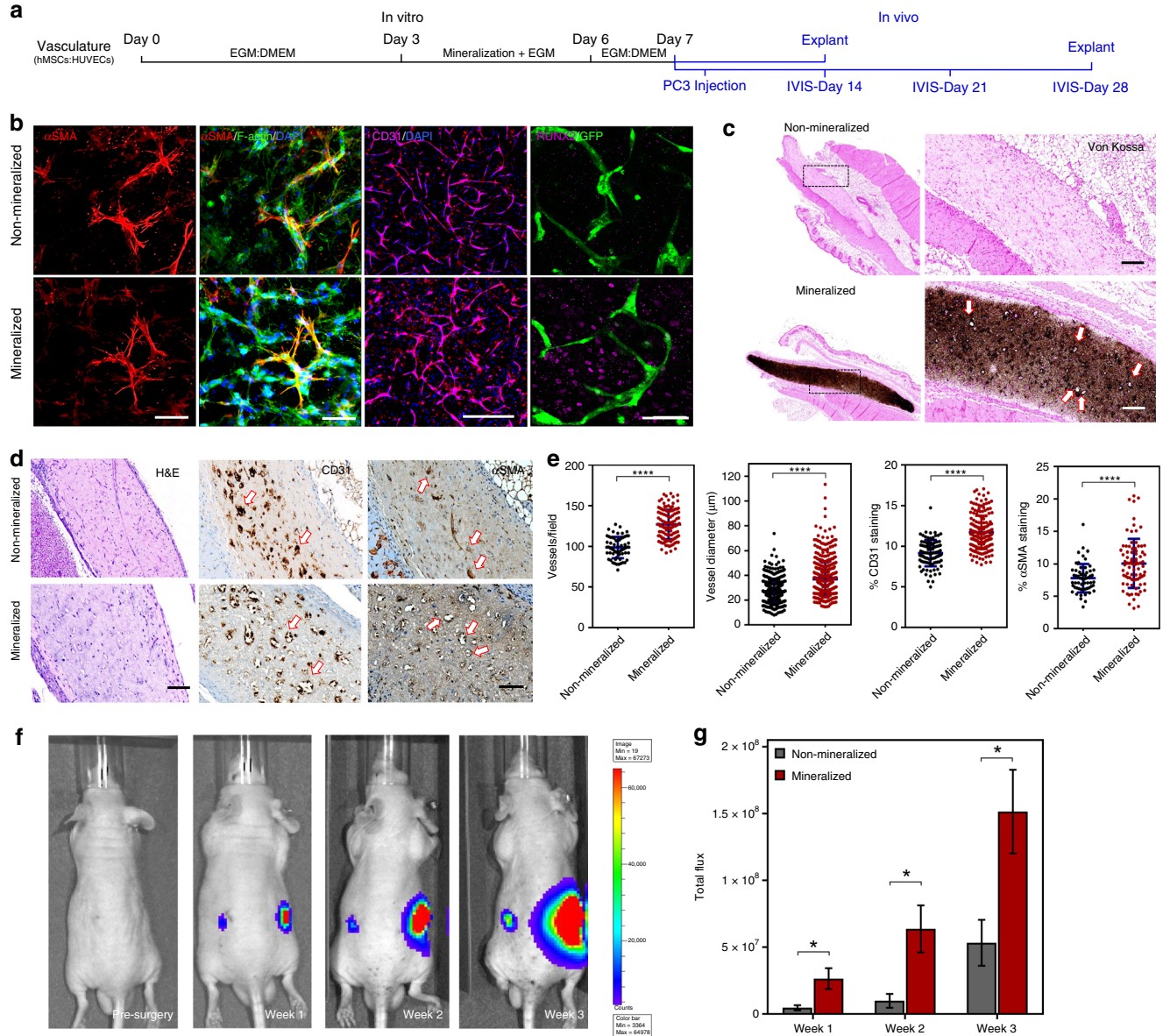

**Fig. 6** Vascularization of mineralized cell-laden collagen and interaction with prostate cancer cells. **a** Timeline for culture and implantation in SCID mice. **b** HUVECs formed endothelial networks that were supported by αSMA-expressing hMSCs (scale bar: 50 μm) and were also positive for CD31 (scale bar: 400 μm). The remainder of hMSCs expressed RUNX2 as a marker for osteogenic differentiation (scale bar: 50 μm). **c** Von Kossa staining of the non-mineralized (top) vs. mineralized tissue sections (bottom) after 7 days of in-vitro culture and 7 days of implantation. The dark/brown staining indicates calcification. Higher-magnification images (right) show luminal structures (arrows) consistent with microvessel formation within the calcified construct. **d** H&E images depict the collagenous matrix populated with cells. Anti-human CD31 antibody staining (middle) suggests the formation of endothelial networks by the transplanted HUVECs as opposed to murine vasculature infiltration. Vessels in non-mineralized sections had constricted lumens (arrows), as opposed to wider HUVEC-lined vessel structures in the mineralized construct (arrows). Anti-αSMA staining (right) shows fewer αSMA+ cells in the non-mineralized sections than in mineralized constructs, which appear to be wrapped by pericyte-like cells. **e** Quantification of vessel number and diameter indicate robust vascularization and cell survival in mineralized groups, compared with their non-mineralized controls. Quantitative analysis of % area of CD31 and αSMA immunostaining suggests an increased vascularization and vessel stability in mineralized constructs. Data presented as Mean ± SD ($N = 3$) (****$p < 0.0001$ Student's $t$-test). **f** Representative bioluminescence images captured before and once a week until 3 weeks, after injection of PC3/Luc cells as visualized by IVIS. **g** A significantly higher bioluminescence signal intensity was detected in the region implanted with mineralized construct compared with the non-mineralized control. Data presented as Mean ± SEM (two-tailed Student's $t$-test, *$p < 0.05$; $n = 8$ for non-mineralized groups; $n = 12$ for mineralized groups from two-independent experiments)

mice. Histological examination of the mineralized vs. non-mineralized constructs explanted after 7 days of implantation showed a high survival of transplanted human cells within both engineered constructs (Supplementary Fig. 21). Von Kossa staining revealed intense dark calcium deposits homogenously distributed throughout the mineralized constructs, whereas no

mineralization was detected in non-mineralized controls (Fig. 6c). Samples were immunohistochemically stained against human-specific CD31 antibody and showed tight endothelial cell junctions that are characteristic of functional blood vessels, indicating that those vessels were formed by the implanted human endothelial cells and not by the invading host vessels

(Fig. 6d). Interestingly, despite a significant reduction in vessel quantity and size after 7 days in vitro (Supplementary Fig. 20), quantification of the microvessel density revealed a significantly higher number of CD31+ vessels in the mineralized constructs compared with non-mineralized controls in vivo (Fig. 6e). Mineralized constructs also developed much larger sized microvessels than non-mineralized samples. Likewise, the percentage areas of CD31+ and αSMA+ staining (Fig. 6e) were also significantly higher in mineralized constructs when compared with the non-mineralized controls.

Lastly, to demonstrate the efficacy of our engineered construct as a disease model system, the mineralized 3D constructs were implanted into the subcutaneous pockets of immunocompromised mice to create ectopic bone-like microenvironments. It is known that the paracrine signals exerted by the cells in the native bone, as well as matrix molecules released by the actively remodeling mineralized tissue, act as crucial players in providing a conducive environment for the proliferation of disseminated prostate cancer cells. To replicate this phenomenon, 24 h post implantation of the cell-laden tissue constructs, a suspension of luciferase-expressing prostate cancer cells (PC3) was injected directly at the ectopic site, so as to determine the ability of our mineralized bone model to condition the growth and colonization of tumor cells. In vivo bioluminescence imaging was performed over a period of 3 weeks to track the outgrowth of PC3 cells. Tumor growth kinetics was significantly higher in mineralized samples than in non-mineralized controls at all time points (Fig. 6f, g). Of note, the target site implanted with the mineralized construct emitted almost threefold higher bioluminescence signal than the non-mineralized counterparts at the end of 3 weeks. Overall, the results implied that the vascularized and mineralized bone-like tissue constructs emulate the grafting and growth of prostate cancer cells in relation to their proximity to the bone. These results illustrate the ability of the proposed model system to study bone-targeting diseases, such as various cancers, with greater experimental control, which should allow for improved screening of therapeutic interventions targeting the interaction between cancers and bone.

## Discussion

Significant efforts have been expended towards engineering bone-like tissue constructs in recent years. Despite substantial progress, there have been no strategies that enable culture of osteoprogenitor, vascular, and neural cells (or other cell types) embedded in a matrix that is controllably calcified on the nanoscale, which is a fundamental characteristic of the human bone microenvironment. Here we demonstrate that a cell-laden collagen hydrogel can be mineralized to mimic the intra- and extrafibrillar nanoscale mineralization profile of native bone, and that such a microenvironment alone is sufficient to stimulate the osteogenic differentiation of hMSCs, while also enabling the formation of hMSC-supported vascular capillaries in vitro and in vivo. Different from traditional osteogenic cell culture protocols, where cells begin to secrete small and dispersed mineral nodules after 14–21 days of culture[33], our results show that the current approach, which uses a supersaturated calcium and phosphate containing medium stabilized by the presence of mOPN, enables widespread and nearly homogenous (Figs. 3m and 4) mineralization of hMSC-laden collagen hydrogels having ultrastructural organization (Fig. 1b, e, f), elemental composition (Fig. 1c), crystallinity (Fig. 1g), and mineral-to-matrix ratio (Fig. 1i), which are comparable to that of human bone in as little as 72 h. Further, our nanomechanical assessment of individual fibrils (Fig. 1k) were consistent with previous reports of nanoindentation modulus of non-mineralized collagen[50] and mineralized collagen

extracted from tooth dentin[51]—which has a nearly identical mineralization profile to native bone—and also closely approximate the values reported for mineralized bone collagen measured using AFM force spectroscopy (pulling)[52]. Nevertheless, there are still important compositional, structural, and mechanical characteristics that would need to be optimized to fully mimic the expected load-bearing function of mature human bone. These form the basis for future studies utilizing the process that we describe here.

In our engineered bone-like tissue constructs, the regions adjacent to the embedded cells appeared more densely compacted with mineral clusters when compared with the regions farther away from the cells (Fig. 4), hinting at active matrix remodeling. Moreover, the cell bodies generated numerous processes that frequently protruded and retracted, indicating the effort to either probe or physically deform the surrounding mineralized matrix (Supplementary Fig. 17). In fact, despite the increased local stiffness (Fig. 1k) and density of the hydrogel due to the formation of mineral crystallites, our results indicate that cells continued to spread, exhibiting numerous dendritic extensions that have strikingly similar morphology to maturing osteocytes[53] (Fig. 3p and Supplementary Fig. 15). A similar observation was noticed by Mata et al.[54], who described increased cell processes and osteoblastic differentiation in hole microtextures as opposed to groove microtextures, likely due to cells sensing these features as enclosed microenvironments with close cell–cell and cell–ECM contact. The formation of dendrite-like protrusions by cells could indicate a shift from a proteolytically degradation-driven process of cell spreading[55], to a scenario where cells have to squeeze through the inter-fibrillar spaces between the stiffened fibers to make their way toward adjacent cells and establish cell–cell communication. This is consistent with previous reports that the formation of osteocyte processes is mainly triggered by intercellular separation and ECM mechanics[56]. Collectively, these observations may point to relevant morphological changes in stem cells as they respond to the gradually stiffening and calcifying bone microenvironment, which characterizes the transition from osteoid to mature and more heavily mineralized tissue. Overall, our results suggest that hMSCs expressed a multitude of characteristics that are consistent with maturing bone cells, which were driven primarily by their encapsulation in a microenvironment that closely mimics that of the native human bone; a set of observations that, to the best of our knowledge, have not been reported previously.

It has been well characterized that hMSCs are sensitive to a diverse array of microenvironmental cues[38]. Two factors that are known to influence stem cell differentiation in our proposed system are matrix stiffness and the presence of calcium and phosphate ions[38,57,58]. The osteoinductive nature of calcium and phosphate scaffolds has long been attributed to their capability of modulating extracellular concentrations of ionized Ca and P, which are sensed by cells either via Ras/Raf/ERK-dependent signaling pathways[58] or ATP-adenosine-controlled mechanisms[57]. Our results showed that cell-laden mineralized hydrogels lead to a marked upregulation of several osteogenic genes (RUNX2, OCN, PDPN, and DMP1) in comparison with non-mineralized controls (Fig. 3a). Interestingly, evidence suggests that calcium and phosphate substrates alone can be as potent as dexamethasone-supplemented medium in inducing osteogenic differentiation[39], which is also consistent with our gene expression data showing comparable or higher levels of mRNA between our mineralized constructs and the positive controls using OIM. Supplementation of calcium and phosphate to the cell-laden constructs alone, however, did not have a significant effect on the expression of differentiation genes (Supplementary Fig. 10), which suggests that bone-like apatite formation may be a

pre-requisite for differentiation of hMSCs embedded in hydrogels cultured in the absence of growth factors and/or osteoinductive supplements. On the other hand, mOPN alone had a significant effect in increasing the expression of some osteogenic markers, which has been well reported in the literature[40]. However, the expression levels were never as high as in the mineralized samples or positive control, and such an increase was mostly temporary, given that mineralization was absent even after 21 days (Supplementary Fig. 11).

A key characteristic of native bone is the ability of resident cells, especially osteocytes, to regulate tissue homeostasis and remodeling in a paracrine fashion[30,44]. Our results indicate that the hMSCs embedded in mineralized microenvironments secrete significantly higher amounts of BMP-2 and BMP-6 compared with collagen alone (Fig. 3b). Also, as osteocytes play a significant role in osteoclast differentiation and recruitment, these cells are a major source of RANKL, a critical mediator of osteoclastogenesis[46]. During bone remodeling, an upregulation of RANKL is typically associated with a lower expression of OPG[47]. Such a ratio was greatly increased in our mineralized samples as compared with that in OIM-treated samples, thus demonstrating the ability of cells encapsulated in the mineralized matrix in potentially instructing osteoclastogenesis and active matrix remodeling in a paracrine manner.

Lastly, in addition to creating a 3D microenvironment that shares the key hallmarks of native bone extracellular matrix and inherent osteogenic potential, the proposed strategy successfully enables the recapitulation of the formation of hMSC-supported vascular capillaries and innervation prior to the onset of matrix mineralization (Figs. 5, 6b), which is an important step toward the formation of functionally vascularized and innervated bone tissue models. Interestingly, the density and stability of these pre-mineralization engineered vessels was improved following in vivo implantation (Fig. 6e). One compelling view in the literature is the influence of hypoxia in regulating vessel integrity and tubulogenesis of endothelial cells by activating hypoxia-inducible factor[59], which may explain why hMSC-supported vascular capillaries were more stable and denser in mineralized samples than in non-mineralized controls after implantation. Overall, these observations point to an important characteristic of the proposed approach, where the ability to direct extracellular matrix mineralization in a controllable fashion, as shown in Supplementary Fig. 19, allows one to stimulate the formation of hMSC-supported vasculature and innervation prior to the onset of mineral formation, without compromising the function of the formed vessels in vivo. We are unaware of other strategies that enable such a controlled phenomenon.

In summary, we have developed a biomimetic approach for the in vitro engineering of a bone-like model system that replicates the nanoscale mineralization of 3D bone microenvironments loaded with osteoprogenitor, vascular, and neural cells[33]. Our results show that the proposed approach enables widespread and nearly homogenous mineralization of hMSC-laden collagen hydrogels having ultrastructural organization and elemental composition that are comparable to that of human bone. The approach is also time-controllable, with the versatility of the synthesis being initiated and stopped at different time points (Supplementary Fig. 19), and being highly cytocompatible. We propose that this model may have important implications for ongoing efforts towards drug discovery and screening, regenerative medicine, and the understanding of bone physiology and disease.

## Methods

**Cell culture**. All experiments used mesenchymal stem cells isolated from human bone marrow. Cells were used from passages 2 to 4. Prior to experiments, cells were cultured in DMEM with 10% fetal bovine serum (FBS), 1% L-Glutamine (200 mM), and 1% antibiotic solution. Similarly, HUVECs expressing GFP (Angioproteomie, cAP-0001GFP) were cultured in Endothelial Growth Media (EGM-Lonza, CC-3162). SH-SY5Y neuroblastoma cells (ATCC, CRL-2266) were cultured in growth medium containing a mixture of DMEM and Ham's F-12 medium (1:1) supplemented with 10% FBS, 1% L-Glutamine (200 mM), and 1% antibiotic solution. Cells were maintained in culture flasks at 37 °C in a humidified atmosphere containing 5% CO$_2$ in air and sub-cultured using 0.25% trypsin-EDTA when cells reached 80–90% confluency.

**Cell-laden hydrogels**. To prepare cell-laden collagen hydrogels, acid solubilized Type 1 collagen from rat tail tendon (3 mg/mL, BD Biosciences) was reconstituted in an ice bath to a final concentration of 1.5 mg/mL in 10× phosphate-buffered saline (PBS) along with DMEM containing a hMSC suspension of $5 \times 10^5$ cells/mL. The pH was adjusted to 7.4 by neutralizing the hydrogel precursors with 1 N NaOH. One hundred microliters of the gels were pipetted onto 24-well plates and were allowed to undergo fibrillogenesis in a humidified 5% CO$_2$ incubator at 37 °C for 30 min. For pericyte-supported endothelial tubulogenesis, hUVECs and hMSCs were encapsulated in collagen at a ratio of 4:1 to a final concentration of $2.5 \times 10^6$ cells/mL, cultured for 3 days, and the constructs were then mineralized. Likewise, for neurogenic induction, SH-SY5Y cells co-encapsulated with hMSCs (4:1) were pre-differentiated with 10 μM retinoic acid (RA) containing low serum medium for 7 days, followed by differentiation in neurobasal medium supplemented with a combination of B-27 supplement, 10 μM RA, 50 ng/mL brain-derived neurotrophic factor (BDNF), 1% FBS, 1% L-Glutamine (200 mM), and 1% antibiotic solution, for additional 7 days, after which the constructs were mineralized.

**Nanoscale hydrogel mineralization**. In order to induce mineralization of collagen in the presence of cells, a modified mineralization medium was formulated by mixing equal volumes of 9 mM CaCl$_2$·2H$_2$O (J.T. Baker) and 4.2 mM K$_2$HPO$_4$ (J.T. Baker) in DMEM supplemented with 10% FBS. Osteopontin powder, extracted from bovine milk (Lacprodan® OPN-10; Arla Foods Ingredients Group P/S, Denmark), was used at a concentration of 100 μg/mL to serve as the mineralization-directing agent and was added in the CaCl$_2$ containing medium before the addition of K$_2$HPO$_4$. To ensure stable maintenance of pH at 7.4, 25 mM HEPES was added to the medium. The samples were incubated under continuous agitation in a rotary shaker so as to ensure uniform mineralization throughout the samples. The mineralizing medium was replenished every 24 h for the first 3 days to induce complete calcification of the collagen gels. Subsequently, constructs were cultured using DMEM with 10% FBS without mineralization supplements for the rest of the culture period. Cell culture medium supplemented with a cocktail of osteoinductive factors containing dexamethasone (100 nM), ascorbic acid (50 μM), and β-glycerol phosphate (10 mM) were used as a positive control (denoted as OIM). For vascularization experiments involving co-culture of hUVECs with hMSCs, samples were cultured in DMEM-EGM-2 medium for 3 days, after which a mineralizing medium supplemented with EGM-2 Bullet Kit was used as described before. Alternatively, for innervation experiments involving co-culture of SH-SY5Y with hMSCs, the cells were subjected to neurogenic differentiation for 14 days, followed by 3 days exposure to mineralizing medium supplemented with a mixture of B-27 supplement, 10 μM RA, and 50 ng/mL BDNF.

**Scanning electron microscopy**. For SEM analysis, samples were fixed with 2.5% glutaraldehyde for 1 h at room temperature, washed in distilled water, and subjected to a series of ethanol dehydration steps for 10 min each. Subsequently, the samples were critical point dried, sputter coated with gold/palladium, and observed under SEM (FEI Helios Nanolab™ 660 DualBeam™) ($N = 6$). The elemental analysis for the presence of Ca and P was carried out using the attached EDX detector (INCA, Oxford Instruments) ($N = 4$).

**Transmission electron microscopy**. For TEM imaging, both mineralized and non-mineralized hydrogels were minced with a double-edge razor blade and were immersed in ice-cold 0.1 M ammonium bicarbonate (pH 7.8). While on ice, the minced hydrogels were then exposed to the cutting blades of an OMNI 2000 tissue homogenizer (OMNI International, Kennesaw, GA) operated at ~11,700 × g until no visible fragments remained. The homogenate was then pipetted onto freshly glow-discharged 600-mesh carbon-coated TEM grids and observed directly using FEI G20 TEM operated at 120 kV ($N = 3$).

For tilt-series electron tomography analysis, the homogenized hydrogels were exposed overnight at 4 °C to 1.5% glutaraldehyde/1.5% formaldehyde with 0.05% tannic acid, then dehydrated and embedded in Spurrs epoxy. Following, 450 nm-thick sections were cut with a diamond knife using a Leica EM UC7 ultramicrotome and mounted on formvar coated 1 × 2 mm slot grids. Sections were subsequently stained in uranyl acetate and lead citrate, and imaged at 200 kV using FEI G20 TEM. For 3D tilt series, 450 nm-thick sections were imaged at 2° increments between 0° and 40°, then at 1° increments between 40° and 70°, then identically imaged from 0° to −40° and −40° to −70°. Tilt series images were collected using FEI Eagle camera directed by FEI Tomography software, then aligned using FEI "Inspect 3D" software ($N = 6$). For SAED analysis, samples were freeze dried in liquid nitrogen and placed between two lacey carbon TEM grids and

imaged using a TECNAI F20 TEM with an Oxford SDD EDS detector and Gatan GIF 2001 system operated at 200 kV ($N = 4$).

**FTIR analysis**. FTIR spectra were obtained in transmission mode (Nicolet 6700, Thermo Scientific) using 32 scans in the range of 4000–400 $cm^{-1}$ at a resolution of 4 $cm^{-1}$. The mineral to matrix ratio was calculated from the area of $v^3PO4$ (1030 $cm^{-1}$) over amide (1660 $cm^{-1}$) peaks after baseline correction and normalization. The crystallinity index was calculated from the parameter splitting factor corresponding to the doublet peak in the fingerprint region (500–650 $cm^{-1}$) that is attributed mainly to $v_4PO_4^{3-}$ bending vibrations. The parameter is calculated as the sum of the peak heights at 565 $cm^{-1}$ and 605 $cm^{-1}$ divided by the height of the minimum between this doublet at 590 $cm^{-1}$. All the height measurements were performed using Origin 8.0 software after baseline correction and normalization of the spectra to the intensity of amide I band (1585–1720$cm^{-1}$) ($N = 6$).

**Atomic force microscopy**. The nanomechanical properties of individual non-mineralized and mineralized collagen fibrils were investigated using a Nanoscope 8 atomic force microscope (J scanner, Bruker) in PeakForce tapping mode. The indentation measurements were performed both in the hydrated state and in air. Al-coated, silicon AFM tips of 300 kHz resonance frequency, 26 N/m nominal spring constant, and a tip curvature radius of ~10 nm (AC160TS; Olympus) were used for non-mineralized collagen fibril tested in air and for mineralized collagen fibril measured in air and in water. On the other hand, Au-coated $Si_3N_4$ AFM tips of 65 kHz resonance frequency, 0.35 N/m nominal spring constant, and a tip curvature radius of ~30 nm (DNP-S, triangle A, Bruker) were used for non-mineralized collagen fibrils in water. These specific cantilevers were chosen to match the stiffness of collagen or mineralized collagen for optimizing the sensitivity. The spring constant of the cantilever was calibrated by the thermal tuning method[56]. After acquiring the two-dimensional topographic image of the fibril, the load-displacement curves at 5–12 randomly selected spots on mica and on the fibril selected were collected under quasi-static indentations. The loading-unloading rate was set to be 100 nm/s, with zero delays in-between. The elastic modulus was obtained by performing Hertz fits of the indentation force against depth curves, as described previously ($N = 3$)[57–59].

**Live and dead assay**. The viability of the cells encapsulated in hydrogels was determined using a live and dead assay kit (Molecular Probes). Cells were incubated for 10 min, followed by rinsing in PBS and imaging using an inverted fluorescence microscope (FL Auto, Evos). Live and dead cell numbers were counted using ImageJ and the percentage of viable cells was quantified as the number of live cells divided by the total cell number ($N = 6$).

**Reactive oxidative stress**. ROS were measured using a CM-H2DCFDA (Abcam) kit to detect any cellular oxidative damage during the mineralization procedure. Measurements were performed immediately after mineralization and after 7 days of culture. Briefly, cell-laden hydrogels were stained in culture media with 20 μM 2',7' –dichlorofluorescin diacetate (DCFDA) for 30 min at 37 °C. The 2', 7' –dichlorofluorescein (DCF) fluorescence intensity was measured using a fluorescence microplate reader with excitation and emission at 485 nm and 535 nm, respectively. Tert-Butyl Hydrogen Peroxide was used as the positive control for detection of ROS ($N = 6$).

**Proliferation assay**. hMSCs (5000 cells per hydrogel) were encapsulated within each non-mineralized, mineralized, and OIM-treated hydrogels, and were cultured for durations of 1, 3, 7, 14, and 21 days. At the end of each of these time points, the culture medium was replaced with fresh medium containing 10% v/v AlamarBlue and the cells were allowed to incubate for 5 h. Subsequently, the formation of fluorescent resazurin products in aliquots of the culture medium was measured in 96-well plates using Tecan Infinite M200 Pro microplate reader (Tecan Trading AG) at excitation and emission wavelength of 550 nm and 590 nm, respectively. The fluorescent readings were then correlated to the cell number by plotting a standard curve of known cell numbers over a range of $5 \times 10^3$ to $8 \times 10^4$ ($N = 6$).

**Serial block face SEM**. For Serial Block Face-SEM, the samples were fixed in Karnovsky's fixative overnight, followed by microwave-assisted embedding process using BioWave Pelco Microwave. Briefly, after washing with 0.1 M cacodylate buffer, the samples were successively post-fixed in 1% osmium tetroxide containing 1.5% potassium ferrocyanide in 0.1 M cacodylate buffer and then immersed in 1% tannic acid, followed by 2% aqueous osmium tetroxide, and finally staining in 1% aqueous uranyl acetate. The samples were then rinsed, dehydrated with a graded series of acetone, and were subsequently embedded in Epon resin. The resin-embedded samples were sputter coated with platinum/palladium. A series of block-face images were obtained using a scanning electron microscope (Teneo Volumescope™, FEI) equipped with an in-chamber ultramicrotome ($N = 3$). A sequence of images was acquired every 60 nm depth with a backscattered electron detector at an acceleration voltage of 2.7 kV under high vacuum. Selected serial thin-section images were then loaded into an image analyses software (Amira) and processed using a 3D reconstruction plug-in (DualBeam 3D Wizard). The

segmentation of cells was done by manually outlining the cell borders, whereas the high contrast difference between the mineral and the non-mineralized collagen was distinguished using a threshold tool. The segmented data sets were further volume-rendered and animated using the Amira Animation Director tool ($N = 3$).

**Confocal microscopy**. A laser-scanning confocal microscope (Zeiss LSM 880) was used for immunofluorescence and reflectance imaging. Briefly, for the immuno-fluorescent staining, samples were fixed with 4% paraformaldehyde and permeabilized using 0.1% Triton X-100 ($N = 3$). The constructs were further blocked using 1.5% bovine serum albumin in PBS for 1 h, followed by incubation with Image-iT FX signal enhancer (Invitrogen, CA) for 30 min to remove background staining. Cells were then incubated with primary antibodies overnight at 4 °C, as listed below. Subsequently, cells were washed three times with PBS/0.1% Tween-20 and incubated with secondary antibodies overnight at 4 °C. The following primary antibodies were used: rabbit polyclonal anti-OCN (Bioss antibodies, bs4917R) (1:50 dilution), mouse monoclonal anti-PDPN (Origene, DM3500P) (1:100 dilution), Rabbit polyclonal anti-DMP1 (Invitrogen, PA5-57956) (1:200 dilution), mouse monoclonal anti-CD31 (Dako, JC70A) (1:200 dilution), rabbit polyclonal anti-RUNX2/CBFA1 antibody (Novus Biologicals, NBP1-77461) (1:100 dilution), mouse monoclonal anti-αSMA (Invitrogen, MA5-11547) (1:400 dilution), mouse monoclonal anti-NEFL antibody (Thermo Fisher Scientific, MA1-2010) (1:50 dilution), and mouse monoclonal anti-Neuron-specific enolase antibody (Abcam, ab218388) (1:1000 dilution). The following secondary antibodies were used at the specified dilutions: Alexa Flour 555 goat anti-mouse IgG (Thermo Fisher Scientific, A21422) (1:200 dilution) and Alexa Fluor 647 goat anti-rabbit IgG (Thermo Fisher Scientific, A21244) (1:200 dilution).

The F-actin was visualized by staining with Alexa Fluor 488-conjugated phalloidin and the nucleus was stained with 4′,6-diamidino-2-phenylindole. For reflectance imaging of collagen fibrils ($N = 4$), the microscope was configured to capture the reflected light between 485 nm and 495 nm, after exciting with a 514 nm laser. The 3D reconstructions of z-stacks of samples were processed and rendered on ZEN black (Zeiss) and Imaris 8 (Bitplane) software. For the quantification of vessel parameters, the images were analyzed using AngioTool (National Cancer Institute, NIH). Likewise, neuronal morphometric analysis was performed using Imaris Filament Tracer module.

**Real-time PCR**. Total RNA from hMSCs were isolated using Tri reagent (Zymogen, USA) according to the manufacturer's instructions. After determining the purity and concentration of the extracted RNA by Nanodrop (Thermo Scientific, USA), complementary DNA was reverse transcribed from 1 μg RNA using SuperScript III first-strand synthesis system (Invitrogen). Quantitative PCR was performed using Power SYBR® Green PCR Master Mix with the following cycling conditions: pre-incubation at 95 °C for 10 min; 40 cycles of denaturation at 95 °C for 30 s, annealing at 50–60 °C for 30 s; and extension at 95 °C for 30 s, followed by met curve analysis to validate the specificity of PCR products. The sequences of the primer set used for the study is provided in Supplementary Table 1. The specified primers were designed with Primer 3 software and blasted against GenBank database sequences to achieve high-specificity primers. GAPDH was used as the internal reference gene for normalization. The fold change in the expression of each gene was calculated using the $2^{-\Delta\Delta ct}$ method ($N = 3$).

**Antibody array profiling for protein expression**. Bone metabolism and remodeling-associated proteins/cytokines were quantified using a multiplex enzyme-linked immunosorbent assayarray (Human Bone Metabolism Array Q2; Raybiotech), according to the manufacturer's instructions. Proteins were isolated either from cell lysates or conditioned medium after 14 days and stored at −80 °C until use. Subsequently, samples were incubated in an array chip printed with capture antibodies of interest. The chips were then incubated with biotinylated detection antibody cocktail, followed by incubation in Cy3-labeled Streptavidin. The slides were then scanned using a gene microarray laser scanner and the signal intensities were detected using densitometric analysis to semi-quantitatively measure the protein level ($N = 5$).

**In vivo implantation**. Subcutaneous implantation was performed in 5–7 weeks old, female SCID beige mice (Charles River Laboratories), after approval from the institutional animal research ethics committee. The samples were randomized and implanted into four separate sites on the back of each mouse ($N = 6$). One week after implantation, the samples were removed, fixed in 10% neutral buffered formalin, and embedded in paraffin. The embedded samples were then sectioned (5 μm thick) and stained using hematoxylin and eosin, von Kossa, Masson's Trichrome, human-specific CD31 monoclonal antibody (1:250; company name), anti-αSMA (1:800; company name, recognizes both mouse and human αSMA) and Rabbit polyclonal anti-DMP1 (Invitrogen) (1:100 dilution). Secondary antibody staining was performed using horseradish peroxidase-conjugated anti-rabbit/ mouse IgG antibody and the peroxidase activity was detected using the 3,3-diaminobenzidine detection system. All the immunostained sections were counterstained with Mayer's hematoxylin (Sigma-Aldrich). The whole slides were then digitized using a Zeiss AxioScan Z1 Slide scanner at ×20 objective. Further, the total number of vessels per field were determined by counting the CD31$^+$ vessels

within the construct. Similarly, the diameter of the CD31-positive vessels were manually quantified using ImageJ. The percentage area of CD31$^+$ and αSMA$^+$ staining was quantified using a color deconvolution plug-in, followed by threshold setting and automated quantification of the immunostained area fraction by ImageJ. Mean ± SD values presented for each experimental group correspond to the average values obtained from at least three animals per group.

To study the interaction of prostate cancer cells with our engineered bone model, the constructs were subcutaneously implanted on the left and right dorsal flanks of 6–8 weeks old male athymic mice (two constructs/animal). One day post implantation, luciferase-expressing PC3 cells (ATCC, CRL-1435) (100,000) suspended in 100 μl PBS were injected directly to the target site. The growth rate of PC3 cells was subsequently monitored weekly for up to 3 weeks using an IVIS Spectrum in-vivo imaging system (Perkin Elmer). Mice were given an intraperitoneal injection of 150 mg/kg D-luciferin dissolved in PBS and the emitted luminescence was analyzed using Living Image 4.3 software (Perkin Elmer). The signal intensity expressed as total flux (photons/second) was quantified as the sum of all detected photon flux counts from the region of interest manually drawn around the tumor during data post processing.

**Statistical analysis**. For the experiments involving the comparison of two groups, statistical analysis was performed using two-tailed, unpaired Student's $t$-test (Prism5, GraphPad Software). For experiments involving more than two groups, one-way or two-way analysis of variance with Tukey's post-hoc test for multiple comparisons was used to identify significant differences. A $p$-value < 0.05 was considered statistically significant. All the quantitative data are presented as mean ± SD, unless otherwise stated in the caption.

**Ethics statement**. All animal procedures were performed in strict accordance with the protocols approved by the institutional animal ethics committee of Oregon Health & Science University (IACUC protocol # IP00000570—Subcutaneous implantation study; # IP00000025—Prostate cancer disease model).

**Reporting summary**. Further information on research design is available in the Nature Research Reporting Summary linked to this article.

## Data availability
The source data underlying the main Figs. 1i–k, 2d–f, 3a, b, 5d, 6e, g and Supplementary Figs. 1, 8f, g, 10, 12b, c, 13b, c, 14b, c, 15a, b, 18, and 20 are provided as a Source Data file. All other relevant data supporting the findings of this paper are available from the corresponding author upon reasonable request.

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

## Acknowledgements

Bone marrow-derived human sesenchymal stem cells were kindly donated by Dr Brian Johnstone. We thank Dr Eric Orwoll and Dr Jason Burdick for insight and expertise on bone and stem cell biology, respectively. We also thank Dr C. Lopez, Dr J. Riesterer, E Stempinski, and K Loftis for help with electron microscopy imaging/analysis performed at the Multiscale Microscopy Core (MMC) facility at OHSU. We thank Dr C. Chaw and Dr S. Kaech for the support with confocal imaging/analysis performed at the Advanced Light Microscopy (ALM) Core at OHSU. L.E.B. acknowledges funding from the NIH/ National Institute of Dental and Craniofacial Research (R01DE026170 to L.E.B.), American Academy of Implant Dentistry Foundation, OHSU-PSU Collaboration Project Seed funding and Cancer Early Detection Advanced Research-Knight Cancer Institute. J.T. acknowledges support from the Laboratory Directed Research and Development Program at Pacific Northwest National Laboratory (PNNL). PNNL is a multiprogram national laboratory operated by Battelle for the U.S. Department of Energy (DOE) under Contract DE-AC05-76RL01830. L.G. acknowledges funding from the National Science Foundation under Grant Number (DMR-1309657).

## Author contributions

G.T. and A.A. designed assays, conducted cell experiments, and wrote the manuscript. R.G., L.Z., and R.B. performed surgeries and analyzed data for prostate cancer studies. D.R.K., Z.C., and B.W. conducted electron microscopy and elemental composition analyses, and J.T. conducted the atomic force microscopy analysis. J.M.J. and H.X. performed the surgical procedures in the subcutaneous mouse model. L.G. and J.L.F. assisted with interpretation of results and reviewed the manuscript. L.E.B. conceived the idea, conceptualized and supervised the study, and co-wrote the manuscript.

## Additional information

**Competing interests:** A patent application based on the work described in this manuscript has been filed by L.E.B. under application number 62/864,935. All other authors declare no competing interests.

