## [Peer Review File · Nature Communications]

Reviewers' comments:

Reviewer #1 (Remarks to the Author):

The study presents an ambitious, holistic and complex in vitro model for bone regeneration incorporating cells, mineralization, capillaries, and neural networks. It includes biomineralization in the presence of cells as a cornerstone feature of the model, considers cell entrapment in the process and dissects important events taking place between MSCs and osteocytes within the mineralizing environment. In my opinion it is novel and relevant. However, while the paper has a number of potentially interesting findings, it requires more rigorous data to confirm some of its claims for a publication in a journal of the level of Nature Comm.

The model is a relatively simple system, which is attractive but consequently requires demonstration of a few key parts.

The authors mention some leading approaches aiming to recreate the biomineralization process of bone. Prior to this, it would be helpful to highlight key studies on the use of organic matrixes for triggering and guiding mineralization that recreates elements of biomineralization (alignment, hierarchy, etc). Also, remarkably, the authors do not mention nor discuss seminal papers by Sam Stupp such as <http://science.sciencemag.org/content/294/5547/1684>, <https://onlinelibrary.wiley.com/doi/abs/10.1002/adma.200802242>.

Page 4, please explain the rationale behind the chosen Ca and PO₄ concentration ranges. Also, please state what are typical concentrations in vivo and how your selected ones compare to them.

Page 5,6. "It should be noted that this type of extrafibrillar mineral still appear to be comprised of nanoscopic crystals that are intimately associated with and coating the fibrils, unlike the large spherulitic hydroxyapatite plates that form typically without protein additive". This is an important part of the model and the authors have used FTIR and TEM to characterize it. How was the crystallinity index measured and calculated? Electron diffraction should be conducted to more thoroughly describe presence of crystalline or amorphous material and how this compares to the in vivo scenario. This is a critical point as mineral can be easily grown in many materials so demonstrating that the mineralization taking place in this model recreates that in vivo would be a critical feature of the model. Also, based on the current data, it is not possible to claim that the model replicates the nanoscale mineralization found in in vivo scenarios.

The model relies on a collagen matrix to nucleate and template the mineralization. Correctly, the authors then use this mineralization to examine the stress effect on cell behavior. While this is a reasonable approach, it is not clear how biologically relevant/similar the organic matrix generated here is to the organic matrix in vivo. In other words, the cell stresses generated in vitro may be different to those of the in vivo scenario because of differences in the nano and hierarchical organization of the collagen matrix. Please provide data and discuss to give a sense of how similar (density, fiber alignment, stiffness, thickness, etc) the organic matrix of the model is to the in vivo organic milieu (even if only the collagen part).

Page 8, why was proliferation measured only up to 7 days and not longer?

Page 9, please provide some characterization of the mineralization surrounding the cells (red in Figure 2). Is this crystalline or amorphous? Is this similar to the in vivo scenario for a similar timescale (provide relevant references)?

Page 10, "This indicates that even though approximately 50% of the organic matrix was mineralized (Supplementary Figure S10), cells were still able to pull on and deform the surrounding matrix, hinting to an active remodeling." Please elaborate on the rationale behind this conclusion? Pulling and

deforming the matrix can take place at different stages. Some quantification of matrix degradation or synthesis around the cells should be conducted to make this claim.

Page 11-15. I would suggest for the order of the paper to be changed so that the data is presented first and then speculative observations. First report and discuss the data related to gene and protein expression and then the observations related to morphology and possible osteocyte phenotype. MSCs are versatile cells that have been shown to adopt different morphologies so this part, being before the data, seems too speculative and can be misleading.

Page 11-12. Some discussion and relevant references should be provided regarding the formation of the cell confinement shown on Figure 3. See for example:

<https://pubs.rsc.org/en/content/articlehtml/2009/sm/b819002j>

Figure 5 is too busy and difficult to read. I would suggest to divide this figure in two and make images bigger.

Differences in matrix between mineralized and non-mineralized seem evident. I believe a more thorough characterization of the mineralized matrix should be conducted for the level of robustness required by this journal and because of the central role that this plays in the functioning and potential impact of the model.

Other minor points

- Please ensure all figures have scale bars (exe. Figure S6).
- Figure 4 legend: (A).
- Some subtitles are in capital letters and others not.
- It would be helpful to use arrows or text directly on the images to guide the reader to the important parts of the images in certain Figures such as current Figures 1d, 4c, 5b,c.

Reviewer #2 (Remarks to the Author):

- The major claims of this paper are as follows:
 - o This paper claims to have developed “an approach for on-demand fabrication of bone-like tissue models with unprecedented levels of biomimicry that will have broad implications for disease modelling, drug discovery, and regenerative engineering”.
 - o The authors claim that these constructs “mimic the nanaoscale structure, composition and function of the cell-rich and calcified bone matrix”.
 - o The authors claim to “stimulate osteogenic differentiation of stem cells to levels that are higher than those obtained with osteoinductive supplements”.
 - o The authors claim to have achieved “cell morphology changes and cell-matrix interactions that are unique to the maturation of osteocytes embedded in mineralised bone, and have never been reported before”.
 - o The authors claim to have achieved “the formation of pericyte-supported blood capillaries and integrate neuronal networks that are cemented with a bed of dense minerals”.
- The novelty of the papers lies in the use of Ca, P and mOPN supplements in the cell culture media for in vitro culture of MSCs encapsulated in collagen hydrogels. There has been widespread study of MSCs interacting with collagen hydrogels, of osteocyte differentiation and mineral deposition, but these are not adequately cited (for example studies by O'Brien et al, McGarrigle et al). The claimed novelty is related to the deposition of the mineral along collagen fibres simultaneously with the development of osteocyte-like cells, and providing a matrix appropriate for vascularisation, innervation and cancer cell homing. However, there is insufficient evidence to substantiate these claims, as is outlined further below. This study would be of interest to others in the community if the paper were substantially revised (see below). It is unlikely to be of interest to the wider field.

- The findings and results are original, but the conclusions are overinflated and not substantiated by the results provided for the following reasons:
 - o A serious concern is that the conclusions are all based on very small sample sizes (N=4 for FTIR, ROS and N=3 for SEM, Live-Dead and PCR. No sample sizes mentioned for TEM and serial block SEM). It is not clear whether the experiments were repeated, or these were all done in the same timeline. This not provide confidence in the scientific rigour and repeatability of these observations.
 - o There were no positive osteogenic induction media controls used for any of the in vitro studies of nanostructure, composition and cytotoxicity, morphology or osteocyte-like features (see Figure 1-3). This is important to substantiate how this compares to gold standard approaches, which poorly induce mineralisation when compared to the levels and organisation in vivo. Based on the gene expression studies later, while there are increases for some genes with the mineralisation media (versus OIM), these do not concur with the timelines of FTIR that suggests it is identical to real bone at 7 days (whereas gene expression is lower than OIM at 7 days – which cannot recapitulate real bone).
 - o The authors make major claims regarding the extent of mineralisation, which hang largely on the image provided of Alizarin Red staining (Figure 4c) and very limited FTIR/EDX data and immunostaining. The biggest concern is that Figure 4c is an extremely poor quality image (the image appears to be out of focus and/or poorly processed). Is it possible that the red hue is due to the presence of the CaP in the media, rather than deposition on the matrix? The staining is also not consistent with the gene expression, which shows OCN levels to be higher at day 7 in the osteogenic control (see Figure 4a). Moreover, the OCN immunostaining provided in Figure 4(d) is over-processed and shows no difference between mineralised and osteogenic positive control at day 14. The methods for the EDX studies are not described. Additional quantitative evidence of the increase in intracellular and extracellular mineralisation are required to support the conclusions of this paper (e.g. quantitative calcium assay, additional independent staining).
 - o Further to this point, the authors claim that “Mineralized samples had comparable values to that of native bone” after 7 days. This is simply not possible in terms of time required for the cells to make that mineral, so it can only be assumed that this is deposited from the media, but this has not been clearly articulated in the paper. Moreover, they make this claim on the basis of the mineral:matrix ratio, rather than any quantitative comparison of the total mineral in their construct versus real bone (you could have very little mineral and very little matrix and the mineral:matrix ratio would be similar). The authors have not adequately addressed or provided evidence whether the CaP have merely deposited on the scaffold from the media, or whether it enticed the cells to make a mineralized matrix themselves.
 - o The authors claim that these constructs “mimic the nanaoscale structure, composition and function of the cell-rich and calcified bone matrix”. However, in the absence of any mechanical testing data, it is not possible to support the claim that the constructs can recapitulate the load bearing capacity of bone, which is the primary function of bone and is also critical to the biological behaviour of the cells within the construct. This is a serious limitation. For this tissue construct to be a functional representation of bone, they need to provide evidence that the mineral observed is bound to the matrix, and that there is sufficient mineralisation to enhance the mechanical properties. Indeed, a current limitation of most bone-like models is their failure to induce sufficient mineral or recapitulate the mechanical properties of bone. There is insufficient evidence that the study reported here has overcome these limitations.
 - o The logic and timelines investigated are not clear throughout. Some of the parameters were only investigated for 7 days, and it in other cases it is not clear from which precise timepoint data is being presented (e.g. Figure 2 and 3).
 - o The authors claim that the “pericellular empty space....is reminiscent of lacunae in osteonal bone”. This is not correct, Fig. 3 depicts pericellular empty space, which do not occur in vivo; it has been widely published that the PCM of osteocytes fills the pericellular space (studies by Schaffler, Weinbaum, You, McNamara). It is likely that these empty spaces have arisen due to poor fixation of the tissue. In general the images provide do not provide any confidence that the cells have differentiated in widespread fashion (how many of the cells had dendrites?). Figure 2 and the Supplementary video 1 appears to depict a very tubular and very thick dendrite, which is not reminiscent of an osteocyte. Moreover, no scale is given to allow independent assessment.

o Fig. 5 – there is no evidence provided to confirm whether the MSCs formed osteocytes when they were simultaneously cultured with HUVECS/neuroblastoma cells. DMP1/Sost staining could confirm that there was a bone cell network present simultaneously, but the authors only provided

o The term “on-demand” is overused and misleading. Moreover, there is no evidence of the process being “time-contrallable, with the versatility of the synthesis being initiated and stopped on demand” – where is evidence of stopping the mineralisation process by the cells?

- The level of detail provided is inadequate, in particular regarding the timepoints depicted and scale bars for some of the figures.

- The results provided are not convincing. Further evidence to strengthen the conclusions would require (1) an osteogenic positive control for comparison to the results presented in Figures 1-3, (2) quantitative evidence of the increase in intracellular and extracellular mineralisation are required (e.g. quantitative calcium assay, additional independent staining), (3) mechanical test data to confirm that the mineral deposited was mechanically integrated with the collagen matrix and provided a tissue close to that of bone, (4) evidence of the existence of a functional osteocyte network at the same time as the vascular/neuronal networks are established (DMP1/Sost staining).

- The paper could influence thinking in the field if it were substantially moderated to remove the exaggerated conclusions (for e.g. “unprecedented levels of biomimicry”), in terms of functional mechanical and cellular behaviour and how this compares to real bone tissue.

Reviewers' comments:

Reviewer #1 (Remarks to the Author):

1. The study presents an ambitious, holistic and complex in vitro model for bone regeneration incorporating cells, mineralization, capillaries, and neural networks. It includes biomineralization in the presence of cells as a cornerstone feature of the model, considers cell entrapment in the process and dissects important events taking place between MSCs and osteocytes within the mineralizing environment. In my opinion it is novel and relevant. However, while the paper has a number of potentially interesting findings, it requires more rigorous data to confirm some of its claims for a publication in a journal of the level of Nature Comm.

Response: We appreciate the reviewer's positive assessment of our work and are pleased that the reviewer supports the innovative character and usefulness of our findings. We have addressed each comment individually below, and performed a more rigorous and robust characterization of our data. We have also made the modifications suggested by the reviewers with regards to formatting, sequence, rigor and tone for each of our datasets. These are presented individually below and are highlighted in blue in the main paper.

2. The model is a relatively simple system, which is attractive but consequently requires demonstration of a few key parts. The authors mention some leading approaches aiming to recreate the biomineralization process of bone. Prior to this, it would be helpful to highlight key studies on the use of organic matrixes for triggering and guiding mineralization that recreates elements of biomineralization (alignment, hierarchy, etc). Also, remarkably, the authors do not mention nor discuss seminal papers by Sam Stupp such as <http://science.sciencemag.org/content/294/5547/1684>, <https://onlinelibrary.wiley.com/doi/abs/10.1002/adma.200802242>.

Response: We agree with the reviewer's recommendation and have added additional studies concerning matrix guided mineralization. We also acknowledge the seminal contribution by Stupp et al in this area, and have now broadened the scope of our references to encompass biomimetic processes of biomineralization including some of the work by the Stupp group. The following statements have been added:

Page 2, line 4: *In bone biomineralization, the deposition of apatite crystals inside collagen fibrils has been suggested to be synergistically orchestrated by matrix noncollagenous proteins,^{8,9} the periodic arrangement of the tropocollagen molecules,^{2,10} fibril geometry¹¹ and water.^{12,13}*

Page 2, line 12: *Efforts to controllably mimic the process of nanoscale bone biomineralization in-vitro date back to decades ago, and have had increasing levels of success.^{10,16,18,20-23} These have included the use of poly(amino acids) and synthetic organic polyelectrolytes early on,²⁴ and have more recently explored the use of self-assembling peptide-amphiphiles,^{21,22} and anionic polymer acids to mimic the function of non-collagenous proteins in templating hydroxyapatite growth within collagen fibrils in-vitro.^{10,14,16,18,20,25}*

The following new references were added:

11. Newcomb, C.J., Bitton, R., Velichko, Y.S., Snead, M.L. & Stupp, S.I. The Role of Nanoscale Architecture in Supramolecular Templating of Biomimetic Hydroxyapatite Mineralization. *Small* 8, 2195-2202 (2012).
21. Hartgerink, J.D., Beniash, E. & Stupp, S.I. Self-Assembly and Mineralization of Peptide-Amphiphile Nanofibers. *Science* 294, 1684-1688 (2001).
22. Spoerke, E.D., Anthony, S.G. & Stupp, S.I. Enzyme Directed Templating of Artificial Bone Mineral. *Advanced Materials* 21, 425-430 (2009).
23. Elsharkawy, S. et al. Protein disorder–order interplay to guide the growth of hierarchical mineralized structures. *Nature Communications* 9, 2145 (2018).
24. Stupp, S.I., Mejicano, G.C. & Hanson, J.A. Organoapatites: Materials for artificial bone. II. Hardening reactions and properties. *Journal of Biomedical Materials Research* 27, 289-299 (1993).
25. Thula, T.T. et al. In vitro mineralization of dense collagen substrates: a biomimetic approach toward the development of bone-graft materials. *Acta biomaterialia* 7, 3158-3169 (2011).

3. Page 4, please explain the rationale behind the chosen Ca and PO₄ concentration ranges. Also, please state what are typical concentrations in vivo and how your selected ones compare to them.

Response: Before selecting one particular concentration for the cell studies in our paper, we first screened various concentrations of Ca²⁺, PO₄³⁻ and mOPN with respect to their cytotoxic effects on plated hMSCs (**Supplementary figure S1**) ranging from 1.1-18 mM Ca and 0.5-8.4 PO₄³⁻. Our data suggested that the metabolic activity/viability of cells was only compromised at the highest concentration we tested. We then determined the ability of the solutions with concentrations in the lower end of our range to mineralize the hydrogels via Alizarin Red staining (**Supplementary figure S2**). When samples were treated with either 1.125 mM Ca²⁺, 0.525 mM PO₄³⁻ and 100 µg/mL of mOPN daily for 3 days, or 2 times that ion concentration, virtually no Alizarin Red staining was present (N=6). At a concentration of 4.5 mM Ca²⁺ and 2.1 mM PO₄³⁻, stabilized with 100 µg/mL mOPN, which are somewhat higher than serum levels of Ca²⁺ and PO₄³⁻ in the body, mineralization was visible and consistent throughout the matrix after 3 days. Therefore, we decided to use the lowest possible concentration that still resulted in consistent and rapid mineralization without compromising cell viability. The concentrations of extracellular calcium in the body have been reported to be around 2.5 mM for Ca and 1.5 for PO₄³⁻, but values as high as 40 mM are present near remodeling sites. Therefore, the concentrations we used are slightly higher than those typically found in the body.

We have added the following to the description of the Ca and PO₄³⁻ concentrations used in our study, and also included the following figure in our supplementary data:

Page 5, line 10: *We also determined the rate of mineral formation in collagen hydrogels by comparing values within a lower range of Ca²⁺ and PO₄³⁻ concentrations that were not cytotoxic (Supplementary Figure S2). Within such a range, cell medium supplemented with 4.5 mM Ca²⁺ and 2.1 mM PO₄³⁻, stabilized with 100 µg/mL mOPN, which are somewhat higher than levels of Ca²⁺ and PO₄³⁻ typically reported for extracellular fluids in the body,³⁵ were chosen for the remainder of the experiments due to the greater efficiency of matrix mineralization and lack of cytotoxicity toward hMSCs.*

Supplementary Figure S2: Alizarin red staining for collagen hydrogels mineralized with increasing concentrations of Ca²⁺ and PO₄³⁻. Samples were treated with 1.125 mM Ca²⁺ and 0.525 mM PO₄³⁻ daily for 3 days and were analyzed on days (a) 1, (b) 3 and (c) 7. Virtually no red staining is present even after 7 days. The same pattern of negligible mineralization is seen for samples treated with 2.25 mM Ca²⁺ and 1.05 mM PO₄³⁻ (d-f). For samples cultured using 4.5 mM Ca²⁺ and 2.1 mM PO₄³⁻, a marked increase in Alizarin Red staining is seen from day (g) 1 to (h) day 3, and the red hue is maintained on day 7 (i) (N=6).

4. Page 5,6. “It should be noted that this type of extrafibrillar mineral still appear to be comprised of nanoscopic crystals that are intimately associated with and coating the fibrils, unlike the large spherulitic hydroxyapatite plates that form typically without protein additive”. This is an important part of the model and the authors have used FTIR and TEM to characterize it. How was the crystallinity index measured and calculated?

Response: The crystallinity index of the samples was calculated using the parameter splitting factor as described by Weiner et al [Stephen Weiner, OferBar-Yosef. Journal of Archaeological Science. Volume 17, Issue 2, March 1990, Pages 187-196]. The splitting factor corresponds to the doublet peak in the fingerprint region (500-650 cm⁻¹) of the FTIR spectra that is attributed mainly to the PO₄³⁻ bending vibrations. The parameter is calculated as the sum of the peak heights at 565 cm⁻¹ and 605 cm⁻¹ divided by the height of the minimum between this doublet at 590 cm⁻¹. All the height measurements were performed using Origin 8.0 software after baseline correcting and normalizing the spectra to the intensity of amide I band (1585–1720cm⁻¹). We have added this description to the materials and methods section, and also transferred the crystallinity index

data to the main paper, as shown below. Also, to further characterize the crystallinity of the mineral formed, we have performed additional experiments to determine the selected area diffraction patterns (SAED) of intrafibrillar collagen via TEM, which are discussed in detail in the answer below.

Figure 1. (h) FTIR spectra and respective (i) mineral:matrix ratio (**** $p < 0.0001$, ANOVA/Tukey) and (j) crystallinity index (** $p < 0.001$, ANOVA/Tukey) of mineralized and non-mineralized collagen constructs, dotted lines are a reference values of native bone. (N=6) (k) AFM nanoindentation modulus of non-mineralized and mineralized hydrated collagen fibrils. Mineralization resulted in over a 1000-fold increase in stiffness for individual fibrils (**** $p > 0.0001$, Students t -test) (N=3). Dotted line is a reference value of native mineralized collagen.³⁸

Page 28, line 18: The crystallinity index was calculated from the parameter splitting factor corresponding to the doublet peak in the fingerprint region (500-650 cm^{-1}) that is attributed mainly to $\nu_4\text{PO}_4$ - bending vibrations. The parameter is calculated as the sum of the peak heights at 565 cm^{-1} and 605 cm^{-1} divided by the height of the minimum between this doublet at 590 cm^{-1} . All the height measurements were performed using Origin 8.0 software after baseline correction and normalization of the spectra to the intensity of amide I band (1585–1720 cm^{-1}) (N=6).

5. Electron diffraction should be conducted to more thoroughly describe presence of crystalline or amorphous material and how this compares to the in vivo scenario. This is a critical point as mineral can be easily grown in many materials so demonstrating that the mineralization taking place in this model recreates that in vivo would be a critical feature of the model. Also, based on the current data, it is not possible to claim that the model replicates the nanoscale mineralization found in in vivo scenarios.

Response: We agree with the reviewer that this is a critical component of the proposed model. In addition to the data provided initially, we have now performed selective area electron diffraction (SAED) analyses of the mineralized samples via TEM, which show the typical broad arcs for the (002) plane and overlapping arcs for the (112), (211), and (300) planes forming a ring shape, all of which are consistent with the known hexagonal crystal form of hydroxyapatite [Elliott, J. C.; Dowker, S. E. P. Apatite Structures. In *Advances in X-Ray Analysis*; JCPDS-International Centre for Diffraction Data: Newtown Square, PA, 2002, 45]. These results were also comparable to the crystallinity measured via SAED analyses from native bone tissue and those previously reported for SAED patterns from osteoblast-secreted mineral crystallites, which we have added as a comparison in **Supplementary Figure S4**. In short, these results indicate that the pattern of nanoscale mineralization obtained with our proposed model is consistent with the nanoscale mineralization found in native bone. We have added these observations to our supplementary material, and have added the following discussion to the main paper:

Page 6, Line 9: *Selective Area Electron Diffraction (SAED) analyses of mineralized fibrils also revealed the typical broad arcs for the (002) plane and overlapping arcs for the (112), (211), and (300) planes, all of which are consistent with the known hexagonal crystal form of hydroxyapatite in the native bone and in osteoblast secreted apatite crystals (Supplementary Figure S4).*^{3,36}

Supplementary Figure S4: *Selective Area Electron Diffraction (SAED) analyses of (a) native bone, (b) mineralized collagen hydrogels after 3 days, and (c) osteoblast-secreted mineralized matrix [Adapted from Boonrungsiman et al.³⁵ with permission]. Despite the different intensities, the typical (002) plane and overlapping arcs for the (112), (211), and (300) are seen in all three samples, which is suggestive of the crystalline nature of the hydroxyapatite in all systems.*

6. The model relies on a collagen matrix to nucleate and template the mineralization. Correctly, the authors then use this mineralization to examine the stress effect on cell behavior. While this is a reasonable approach, it is not clear how biologically relevant/similar the organic matrix generated here is to the organic matrix in vivo. In other words, the cell stresses generated in vitro may be different to those of the in vivo scenario because of differences in the nano and hierarchical organization of the collagen matrix. Please provide data and discuss to give a sense of how similar (density, fiber alignment, stiffness, thickness, etc) the organic matrix of the model is to the in vivo organic milieu (even if only the collagen part).

Response: The reviewer is correct that the organic matrix plays a fundamental role in the cell-response in our system. We acknowledge that this is the main limitation of our model. Although we have collected significant evidence that the process of nanoscale mineralization closely approximates that of native bone, and that the relative mineral:matrix ratio also approximates that of the native tissue, we acknowledge that the density of the organic matrix that we use is inferior to that of native bone. Accordingly, our organic matrix is a standard low-density collagen hydrogel from rat-tail tendon, reconstituted at 1.5 mg/mL. Collagen hydrogels have been widely used for cell encapsulation studies, and we acknowledge that these hydrogels do not match the alignment or density of collagen in native human tissues, despite being a widely accepted model for cell studies. Having said that, at the level of individual fibrils, which is particularly relevant to our biomineralization process, our collagen matrix matches the natural pattern of collagen hierarchical self-assembly, resulting from laterally organized individual tropocollagen molecules that assemble to form D-banded fibrillar structures ranging from 20-180 nm in diameter, with peaks in the order of 60 and 120 nm in size. This is consistent with individual collagen fibrils in bone and other mineralized tissues. The size, fibrillar pattern and D-banding periodicity of collagen is illustrated in the figure below, and was also added to our supplementary data (**Supplementary Figure S9**).

Moreover, AFM nanoindentation data from non-mineralized samples shows that the stiffness values of collagen used in this study closely approximates that obtained from non-mineralized collagen reported previously. Accordingly the average elastic modulus of non-mineralized collagen was 1.60 ± 0.32 MPa in water, which approximates the 10-40 MPa previously reported by Balooch et al. from demineralized dentin collagen (J Struct Biol. 2008 Jun;162(3):404-10), and that reported by Grant et al., which ranged from 2-5 MPa (Biophysical Journal Volume 97 December 2009 2985–2992) (**Supplementary Figure S8**). The following figures were added:

Supplementary Figure S8: Analysis of nanoindentation elastic modulus of non-mineralized and mineralized collagen fibrils performed in water (a-c) and in air (d-f) using an atomic force microscope (Nanoscope 8 atomic force microscope, J scanner, Bruker). Representative load-displacement curves on (a) non-mineralized and (b) mineralized collagen fibrils, measured in water. The loading curve is represented in red and the unloading curve is represented in blue. The dotted line corresponds to the indentation on fibrils, while the indentation on the adjacent mica substrate is denoted by the solid line. Inset shows the representative AFM topographic image of individual fibrils fixed onto freshly cleaved muscovite mica disc, with the color scale on the right corresponding to the height. (c) Comparison of average elastic modulus in non-mineralized versus mineralized fibrils, calculated by fitting the Hertz model to the loading curves. An increase of over 1000-fold was recorded in hydrated mineralized fibrils versus non-mineralized controls (**** $p < 0.0001$, Student's *t*-test). Corresponding load-displacement curves and average modulus of non-mineralized and mineralized fibrils in air are shown in (d), (e) and (f), respectively. (f) The nanoscale elastic moduli values in air followed the same trend, with a substantial increase in stiffness in mineralized fibrils compared to its non-mineralized counterparts (**** $p < 0.0001$, Student's *t*-test). In all cases, the maximum penetration depth was set to $\sim 15\%$ of the fibril thickness. In (b-e) the measurements were made using aluminum-coated, silicon microsphere tip of radius, 10 nm, resonance frequency of 300 kHz and a nominal spring constant of 26 N/m whereas for non-mineralized collagen fibril in (a), indentation was performed using triangular gold-coated silicon nitride tip of 30 nm radius, 65 kHz resonance frequency and 0.35 N/m nominal spring constant. Measurements were performed on 5 to 10 different locations from at least 3 fibrils per sample, with a total of 3 samples per condition.

Supplementary Figure S9. a) TEM image illustrating the alignment and banding pattern of collagen fibrils used for cell and mineralization studies. b) Frequency plot of fibril diameter from a total of 96 measurements (N=4). The fibrils diameter was typically in the range of 60-120 nm, with higher frequencies found for 60, 100 and 120 nm diameter fibrils.

7. Page 8, why was proliferation measured only up to 7 days and not longer?

Response: We were originally interested in characterizing the early stage cytotoxicity of our process (days 1-3), and then the ability of cells to maintain or recover metabolic activity for a few days after the mineralization period ended. Our ROS data showed a drop in generation of ROS on day 7 (Figure 2e); also the supplementation of Ca^{2+} and PO_4^{3-} to the cells, which presumably was the more toxic stage, only lasted for 3 days, and cells remained viable for at least 4 days after that. Hence, we did not see a need to extend the analyses in the original data. To account for the reviewer's suggestion, the proliferation study was repeated for up to 21 days and the results are updated in **Figure 2f** of the main paper now. A similar trend to what we reported originally is shown. We found that cells increased metabolic activity and proliferation steadily, and then showed a consistent drop on day 21 for all groups; probably due to the effect of differentiation or growth impairment due to cell-cell contact. The following was added:

Figure 2: Cytocompatibility of the biomineralization process. Representative fluorescent micrographs of live (blue) and dead (green) stained hMSCs embedded in (a) non-mineralized, (b) mineralized and (c) OIM treated hydrogels after 7 days. (d) All groups showed more than 90% cell viability on days 1, 3 and 7, irrespective of the treatment method (N=6) Scale bar: 100 μ m. (e) Mineralization of cell-laden hydrogels resulted in non-significant generation of reactive oxygen species (ROS) compared to H₂O₂ (*p<0.05 and ****p<0.0001, ANOVA/Tukey) after 7 days (N=6). (f) Proliferation rate of hMSCs encapsulated in mineralized hydrogels was comparable to that of cells embedded in non-mineralized and OIM treated hydrogels up to day 3, and was significantly higher than both groups on days 7 and 14 (*p<0.05, **p<0.01, ANOVA/Tukey). After 21 days, a sharp decrease in cell number was recorded in all groups, with a significant reduction observed in non-mineralized (*p<0.05 ANOVA/Tukey) and mineralized (**p<0.01 ANOVA/Tukey) hydrogels in comparison to OIM-treated samples. (N=6)

8. Page 9, please provide some characterization of the mineralization surrounding the cells (red in Figure 2). Is this crystalline or amorphous? Is this similar to the in vivo scenario for a similar timescale (provide relevant references)?

Response: As described in question 5 above, our data (Figure 1e and Supplementary figure S4) suggests that the mineral surrounding the cells is crystalline, given the appearance of the (002) plane and overlapping arcs for the (112), (211), and (300), which are consistent with the presence of crystalline hydroxyapatite. The timescale for mineralization, we contend, is far quicker than what is typically observed in a process of cell-driven mineralization using OIM in-vitro (14-21 days) [Gentleman, E. et al. Comparative materials differences revealed in engineered bone as a function of cell-specific differentiation. Nature Materials 8, 763 (2009)]. Moreover, in OIM-treated cells, the resulting minerals are typically apatite nodules that are loosely dispersed in a matrix, while in-vivo mineralization is more homogeneous and contained to a 'mineralization front'. Our data shows that our method can more closely approximate such homogeneous mineralization fronts with the mOPN-guided process. Furthermore, we argue that it is difficult to ascertain the exact speed of individual collagen mineralization in-vivo. The process of osteoid calcification has been reported to last for about 2 weeks [Carlisle and Fischgrund. Bone Graft and Fusion

Enhancement. Surgical Management of Spinal Deformities. 2009, Pages 433-448], however this is also likely a diffusion-limited process, for which the thickness of the osteoid layer greatly influences the speed of mineral formation. The same can be said about our samples, where the mineral formation into the core of the sample was slower for thicker gels. Therefore, it is difficult to precisely ascertain whether our mineralization process is as fast or slower than the mineralization in vivo. Nevertheless, the process of fabrication of our mineralized bone-like samples is substantially faster than existing models of fabrication of bone-like tissue in the lab.

We have added the following description to the paper:

Page 16, line 26: Of note, these fibrils are mineralized with similar levels of crystallinity as those observed in native bone and in osteoblast secreted minerals (**Supplementary Figure S4-S6**).

Supplementary Figure S4: Selective Area Electron Diffraction (SAED) analyses of (a) native bone, (b) mineralized collagen hydrogels after 3 days, and (c) osteoblast-secreted mineralized matrix [Adapted from Boonrungsiman et al.³⁵ with permission]. Despite the different intensities, the typical (002) plane and overlapping arcs for the (112), (211), and (300) are seen in all three samples, which is suggestive of the crystalline nature of the hydroxyapatite in all systems.

Supplementary Figure S5: (a) TEM image of mineralized collagen (unstained) lyophilized and pulverized in liquid nitrogen. Scale bar: 500 nm (b) Needle shaped apatite crystals occupying both

the intrafibrillar and extrafibrillar space of collagen fibrils can be visualized in the high magnification image (from the white dotted square in a). Note that the continuity of crystal orientation along the long axis of the fibers is disrupted primarily during TEM sample processing that involved crushing and dispersing the collagen fibrils on TEM grids, resulting in short and randomly oriented fibers/crystallites. A segment of the fibril that did not mineralized (faint contrast) is also present. Scale bar: 100 nm (N=3)

Supplementary Figure S6: TEM images showing the progressive mineralization of 3D collagenous matrix from day 1 to day 3. (a) Unstained TEM image depicting partially mineralized fibrils at day 1, with numerous mineral clusters deposited onto the surface of collagen fibrils. The corresponding high magnification image is shown in b. Scale bar: 500 nm (c,d) After 3 days of mineralization, extensive needle-like apatite deposition was observed, both at the intra and extrafibrillar space. Scale bar: 250 nm (N=3)

9. Page 10, "This indicates that even though approximately 50% of the organic matrix was

mineralized (Supplementary Figure S10), cells were still able to pull on and deform the surrounding matrix, hinting to an active remodeling.” Please elaborate on the rationale behind this conclusion? Pulling and deforming the matrix can take place at different stages. Some quantification of matrix degradation or synthesis around the cells should be conducted to make this claim.

Response: In order to address this comment, we attempted to qualitatively monitor the cell induced remodeling of the 3D microenvironment by incorporating fluorogenic dye-quenched substrates (DQTM-collagen I, ThermoFisher Scientific) into the mineralized matrix. This method is generally used for the real time monitoring of cell-mediated proteolytic cleavage of their immediate microenvironment. However, in our case, it was extremely difficult to visualize the emitted fluorescent signals from the DQ substrates upon degradation (data not shown), possibly due to the non-specific interaction with the heavily calcified matrix, resulting in the formation of diffuse background fluorescence throughout the matrix.

As an alternative approach, we performed time-lapse imaging of cells labelled with a cell-tracker embedded in mineralized versus OIM hydrogels after 7 days of culture using confocal microscopy in bright field mode. This has now been included as **Supplementary Videos 3 and 4**, and **Supplementary Figure S17**. Images were acquired at 20X magnification for 90 min at 10-min intervals with a Zeiss Airyscan 880 microscope and processed with ImageJ software. Interestingly, the cells embedded in mineralized gels exhibited a distinct motility pattern and protrusive behavior compared to OIM. In the mineralized samples, the cell bodies generated numerous small processes that frequently protruded and retracted, indicating the effort to either probe or physically deform the surrounding stiff mineralized matrix. Moreover, protrusive filopodia were also noticed at multiple regions, appearing to penetrate through the matrix to establish connection with neighboring cells.

Supplementary Figure S17: Time lapse imaging sequence for hMSCs encapsulated in a mineralized collagen hydrogel (N=3) on day 7. Images were acquired at 20X magnification for 90

min at 10-min intervals with Zeiss airyscan 880 microscope and processed with ImageJ software. Cells were stained using a cell tracker (deep red cell tracker, Invitrogen, 1 ug/ml) and images were generated both in fluorescence and transmission modes, and then merged. A single cell is shown with a white arrow to illustrate cell movement within the mineralized matrix. On the image in (a), the right and lower edges of the cell are bordered with a tangential dotted line to mark the starting point of a cell at time point 0; the lines are not moved across the time points for easier reference. The movement of the cell away from the reference lines facilitates visualization of the movement path of the cell from 0 to 90 min (a-j). A zoomed-in view of a single cell at time points 0 (k), 50 (l) and 90 min (m) shows the morphological changes to the cell cytoplasm and changes in the formation of cell processes that extend and contract in different directions over time. In (l) multiple narrow processes are shown in a cell (yellow arrow), and after 90 min, these processes have either disappeared or moved to another location (green arrows). A video of the time-lapse images is shown in Supplementary Video 3, and a comparison for cells cultured in osteoinductive medium is shown in Supplementary Video 4. Scale bar: 50 μ m.

10. Page 11-15. I would suggest for the order of the paper to be changed so that the data is presented first and then speculative observations. First report and discuss the data related to gene and protein expression and then the observations related to morphology and possible osteocyte phenotype. MSCs are versatile cells that have been shown to adopt different morphologies so this part, being before the data, seems too speculative and can be misleading.

Response: We appreciate this suggestion, and we agree that the speculative data would be better placed in a later section of the paper. To account for these suggested and other changes, we have split figure 1 into 2 figures, one focused on the materials characterization (Figure 1), and another on the cytotoxicity response (Figure 2) of the proposed method. Next, we introduced the gene/protein data analyses (Figure 3), and combined the fluorescence characterization of proteins with that of dendrite-like structure, as to compile all of the fluorescence/protein data into a single figure. The electron microscopy images is now presented after the characterization of gene/protein/cell function in Figure 4, where we also introduced images for cells cultured in non-mineralized collagen and in OIM for 7 days, as controls. Lastly, we also divided figure 5 into 2 separate figures, and reported the effects of mineralization on vasculature and innervation separately. We moved the innervation data to the supplementary file, to give greater emphasis to the vasculature component of the paper, which was more thoroughly characterized. To reflect these changes, we then removed the innervation component from the title, and reworded it to read: ***Rapid fabrication of vascularized cell-laden bone models with biomimetic intrafibrillar collagen mineralization, nanostructure and composition.***

Figure 3 now includes:

Figure 3: Osteogenic lineage commitment, gene and protein expression in mineralized cell-laden collagen. (a) Gene expression analyses of hMSCs cultured in non-mineralized versus mineralized constructs (without osteoinductive supplements), compared to cells cultured in osteoinductive medium (OIM, positive control). Gene expression of osteocalcin (OCN) was significantly higher ($**p < 0.01$) in mineralized collagen than in collagen supplemented with OIM after 21 days. Expression of osteocyte-related genes (dentin matrix protein 1 - DMP1 and podoplanin - PDPN) was also comparable to OIM after 21 days of culture in mineralized collagen, and significantly higher at earlier time points (DMP1, $**p < 0.01$ after 7 days, and $*p < 0.05$ after 14 days). ($N=3$) (b) Relative expression of bone metabolism-related proteins by hMSCs in mineralized collagen versus OIM and non-mineralized controls. A significant increase in the expression of BMP 2 and BMP 6 for cells in both mineralized collagen and in OIM relative to non-mineralized controls is consistent with enhanced bone-specific metabolic activity. A marked increase in the ratio of RANKL/OPG in mineralized samples, however, suggests the stronger potency for cell-mediated bone-remodeling via a paracrine signaling in cell-laden mineralized constructs than in the other groups. ($N=4$) At a cell-surface level, the expression of OCN on day 14 was very low in (c) non-mineralized controls, and significantly higher for both (d) mineralized and (e) OIM-treated samples ($**p < 0.01$) ($N=3$). The expression of PDPN on day 14 was also very low in (f) non-mineralized controls, and significantly higher in (g) mineralized collagen than in the (h) OIM-treated group ($***p < 0.001$). ($N=3$) Similarly, surface expression of DMP1 was comparable for OIM-treated cells ($****p < 0.0001$) and mineralized collagen ($*p < 0.05$), both which were significantly higher than non-mineralized controls ($N=4$). Reflectance confocal microscopy images of F-Actin/DAPI stained hMSCs in both (l) non-mineralized, (m) mineralized, and (n) OIM treated cell-laden collagen, illustrate the dendritic-like extensions of cells after matrix mineralization, reminiscent of an osteocyte-like morphology. Images in (o-q) show the reflectance images referred above but without the cells, where the mineralized matrix (p) appears to form well-defined lacunae-like regions in the locations

where cells resided. Scale bar: 100 μm . Quantification of dendrite-like projections is showing in Supplementary Figure S15. Scale bar: 200 μm . All comparisons used ANOVA/Tukey.

11. Page 11-12. Some discussion and relevant references should be provided regarding the formation of the cell confinement shown on Figure 3. See for example: <https://pubs.rsc.org/en/content/articlehtml/2009/sm/b819002i>

Response: We agree with the reviewers' comment. Accordingly, we have updated the revised version to include a discussion on how our current results compare to the observations by Mata et al., which describes greater filopodial extensions and osteoblastic differentiation in hole microtextures as opposed to groove microtextures, likely due to cells sensing these features as enclosed microenvironments with close cell-cell and cell-ECM contact. Additionally, we have now included the time-lapse images capturing the behavior of cells embedded in mineralized matrix (as a part of addressing the reviewers' previous comments) that provides further understanding of how physical confinement influences the morphology and differentiation of stem cells. These observations are particularly relevant to the bone microenvironment, as the majority of the bone cells are confined in a densely calcified matrix.

The following was added:

Page 14, line 7: *A similar observation was noticed by Mata et al., who described greater filopodial extensions and osteoblastic differentiation in hole microtextures as opposed to groove microtextures, likely due to cells sensing these features as enclosed microenvironments with close cell-cell and cell-ECM contact.⁵¹ In non-mineralized (Figure 3l) or OIM-treated controls (Figure 3n), on the other hand, cells lacked any marked increase in the projection of cell dendrites (Supplementary Figure S15).*

12. Figure 5 is too busy and difficult to read. I would suggest to divide this figure in two and make images bigger.

Response: As we described in question 10, we divided figure 5 into 2 separate figures, to report the effects of mineralization on vasculature and innervation separately. We then moved the innervation data to the supplementary information, to give more emphasis to the vasculature component of the paper. Figure 5 now includes:

Figure 5. Vascularization of mineralized cell-laden collagen, and interaction with prostate cancer cells. a) Timeline for culture of vascularized mineralized bone-like tissue constructs, and subcutaneous implantation in SCID mice. b) HUVECs formed interconnected endothelial networks which were supported by α SMA-expressing hMSCs tightly adhered to forming vessels. Scale bar: 50 μ m. The network structures were also positive for the endothelial cell surface marker CD31 (Scale bar: 400 μ m), while the remainder of hMSCs expressed RUNX2 as a marker for osteogenic differentiation. Scale bar: 50 μ m. c) Von Kossa staining of the non-mineralized (top panel) versus mineralized and vascularized tissue sections (bottom panel) after 7 days of in-vitro culture and 7 days of implantation. The dark/brown staining indicates areas of dense calcification in the mineralized tissue sections, whereas the non-mineralized tissue was negative for Von Kossa staining. Higher magnification of the boxed regions (right) reveal the presence of numerous luminal structures (arrows) in the mineralized sections consistent with microvessel formation within the calcified construct. d) H&E images depict the collagenous matrix populated with cells. Anti-human CD31 antibody staining (middle) suggests the formation of endothelial networks by the transplanted HUVECs in both groups, which are not due to infiltration of host murine cells. The vessels in the non-mineralized sections show signs of regression with more constricted lumens (arrows), as opposed to wider HUVEC-lined vessel structures in the mineralized construct (arrows). Anti- α SMA staining (right) shows fewer α SMA⁺ cells in the non-mineralized sections, whereas most of the vessels in the mineralized construct appear to be wrapped by pericyte-like cells. e) Quantification of the vessel parameters (vessel number and diameter)

*indicate robust vascularization and cell survival in mineralized groups, compared to their non-mineralized controls. Quantitative analysis of % area of CD31 and α SMA immunostaining suggests an increased vascularization and vessel stabilization in mineralized constructs. (N=3) (*** $p < 0.0001$ Student's t-test). g) Local invasion of PC3 cells at the ectopic site implanted with mineralized construct as visualized by IVIS. The non-mineralized and mineralized constructs were subcutaneously implanted on the left and right flanks of immunocompromised mice, followed by direct injection of luciferase expressing PC3 cells (1×10^5 cells), 24 h post implantation. Representative bioluminescence images captured before and once a week until 3 weeks, after injection of PC3/Luc cells is shown. h) A significantly higher bioluminescence signal intensity was detected in the region implanted with mineralized construct compared to the non-mineralized control. Data presented as Mean \pm SEM (Two-tailed Student's t-test, * $p < 0.05$; $n=8$ for non-mineralized groups; $n=12$ for mineralized groups from two-independent experiments).*

13. Differences in matrix between mineralized and non-mineralized seem evident. I believe a more thorough characterization of the mineralized matrix should be conducted for the level of robustness required by this journal and because of the central role that this plays in the functioning and potential impact of the model.

Response: We have addressed this comment by providing a more thorough characterization of matrix mineralization. In short, mineralization is now characterized using 10 different methods: 1. FTIR (Spectra, mineral:matrix ratio, and crystallinity index), 2. EDX, 3. Standard SEM, 4. Serial Block-face backscattered SEM (in 3D and 2D), 5. TEM, 6. SAED, 8. Fluorescent Intracellular and 7. Extracellular calcium, 9. Alizarin red, and 10. Von Kossa staining.

Other minor points

- Please ensure all figures have scale bars (exe. Figure S6).
Scale bars have been added to all figures.

Figure 4 legend: (A).

- Some subtitles are in capital letters and others not.

All subtitles in the main paper were switched to lower case letters.

- It would be helpful to use arrows or text directly on the images to guide the reader to the important parts of the images in certain Figures such as current Figures 1d, 4c, 5b,c.

Arrows were added to figures 4g, 5c, 5e.

Reviewer #2 (Remarks to the Author):

• The major claims of this paper are as follows:

o This paper claims to have developed “an approach for on-demand fabrication of bone-like tissue models with unprecedented levels of biomimicry that will have broad implications for disease modelling, drug discovery, and regenerative engineering”.

o The authors claim that these constructs “mimic the nanaoscale structure, composition and function of the cell-rich and calcified bone matrix”.

o The authors claim to “stimulate osteogenic differentiation of stem cells to levels that are higher than those obtained with osteoinductive supplements”.

o The authors claim to have achieved “cell morphology changes and cell-matrix interactions that are unique to the maturation of osteocytes embedded in mineralised bone, and have never been reported before”.

o The authors claim to have achieved “the formation of pericyte-supported blood capillaries and integrate neuronal networks that are cemented with a bed of dense minerals”.

1) The novelty of the papers lies in the use of Ca, P and mOPN supplements in the cell culture media for in vitro culture of MSCs encapsulated in collagen hydrogels. There has been widespread study of MSCs interacting with collagen hydrogels, of osteocyte differentiation and mineral deposition, but these are not adequately cited (for example studies by O’Brien et al, McGarrigle et al). The claimed novelty is related to the deposition of the mineral along collagen fibres simultaneously with the development of osteocyte-like cells, and providing a matrix appropriate for vascularisation, innervation and cancer cell homing. However, there is insufficient evidence to substantiate these claims, as is outlined further below.

Response: We truly appreciate the reviewer’s assessment of our proposed strategy, and acknowledge the constructive criticism and generally positive evaluation of the results that we reported. We also thank the reviewer for the carefully outlined summary above, and regret that the cornerstone feature of our model may have been unclear. Accordingly, as the reviewer indicated below, it appears as though it was unclear whether collagen mineralization was a result of cell-secreted calcification, as in standard osteogenic differentiation models, or if mineral crystallites were deposited by the ions in the medium.

We attest that our strategy works by supplementing soluble calcium and phosphate to the cell culture medium in ionic conditions that are supersaturated with respect to the formation of hydroxyapatite, and then promoting a protein-induced intrafibrillar collagen mineralization process, which uses milk-extracted osteopontin (mOPN) as an anionic non-collagenous protein analogue as a precipitation inhibitor. Accordingly, this composition inhibits spontaneous precipitation of the calcium and phosphate in the medium, and modulates the non-classical mineralization of hydroxyapatite within the collagen fibrils via infiltration of amorphous calcium and phosphate between tropocollagen molecules in a fibril, followed by their precipitation within the fibril structure. This process mimics bone biomineralization, where the extracellular levels of Ca^{2+} and PO_4^{3-} ions secreted by cells are also supersaturated with respect to hydroxyapatite, and mineral precipitation is also inhibited by the action of anionic non-collagenous proteins, which include OPN, to allow for infiltration and subsequent crystallization of amorphous calcium and phosphate inside the fibrils. Of note, the proposed design of our strategy was set out to allow cells that were initially embedded in a mineral-free matrix material to be rapidly (3 days) embedded within a calcified microenvironment that has the nanostructural features of the native bone matrix. This, we argue, is the key novelty of our approach, since there are no methods to embed cells in a nanoscale-calcified matrix similar to that of native bone, especially resulting in osteogenic differentiation of stem cells in the absence of any growth factor supplement.

To more precisely describe this key feature of our model, we have modified the following:

Abstract: Here we describe a biomimetic approach where a supersaturated calcium and phosphate medium is used in combination with a non-collagenous protein analogue to direct the deposition of nanoscale apatite, both in the intra- and extrafibrillar spaces of collagen embedded with osteoprogenitor, vascular and neural cells.

Introduction:

Page 3, Line 18: *Here, we describe the encapsulation of undifferentiated human mesenchymal stem cells (hMSCs) in 3D microenvironments, cultured in supersaturated calcium and phosphate-rich cell media supplemented with a non-collagenous protein analogue, which directs the formation of nanoscale hydroxyapatite in the interstices of collagen fibrils. This process mimics the nanoscale structure, composition and a set of important biological functions that are characteristic to the cell-rich calcified bone microenvironment. We show that the matrix nanoscale mineralization alone can stimulate the osteogenic differentiation of bone marrow-derived stem cells to levels that are comparable to those obtained with standard osteoinductive media. Moreover, this process leads to cell morphology and cell-matrix interactions that are consistent with the characteristics of pre-osteocytes embedded in mineralized bone.*

Results and discussion:

Page 4, Line 19: *Our strategy works by supplementing soluble Ca^{2+} and PO_4^{3-} to the cell culture medium in ionic conditions that are supersaturated with respect to hydroxyapatite, and then promoting a protein-induced intrafibrillar collagen mineralization process, using milk-extracted osteopontin (mOPN), an anionic protein, as a nucleation inhibitor. This process prevents spontaneous precipitation of calcium and phosphate in the medium, while modulating the non-classical (i.e., amorphous precursor) nanoscale mineralization within the collagen fibrils throughout the cell-laden matrix in a rapid fashion.³⁴*

Per the reviewer's suggestion, we have also cited the following papers regarding the interaction of hMSCs with collagen hydrogels and other bone scaffolds:

Ref 29, O'Brien, F. J. Biomaterials & scaffolds for tissue engineering. *Materials today* **13**, 88-95 (2011).

Ref 27, Murphy, C.M., O'Brien, F.J., Little, D.G. & Schindeler, A. Cell-scaffold interactions in the bone tissue engineering triad. *European cells & materials* **26**, 120-132 (2013).

2) This study would be of interest to others in the community if the paper were substantially revised (see below). It is unlikely to be of interest to the wider field.

Response: We appreciate the reviewer's assessment of the potential interest of others in the field, and have now substantially revised the manuscript. We argue, as well, that given the lack of model systems that replicate the definitive characteristic of bone tissue – which is an organic-inorganic composite material embedded with metabolically active cells cemented within a bed of fibrillar proteins calcified on the nanoscale – this strategy may extend beyond the scope of bone research. For instance, several cancers (i.e. breast, prostate, lung, and others) frequently metastasize into bone, and in-vitro models that enable controlled studies on how cancers invade the mineralized matrix as influenced by host cells are lacking. The function of the bone marrow has also long been acknowledged to be heavily influenced by the endosteal niche, and its embedded cells, and model systems that replicate this feature of the marrow tissue do not currently exist. Similarly, studies addressing the ingrowth of diseases, pathogens, and implants into bony tissue depend on the ability of model systems to replicate the key hallmarks of bone tissue, including its mineralization and the cells embedded within such a calcified matrix. In the same note, the ability of drugs to interact with bone tissue, bind to its matrix, and target its cells, all depend on model systems that truly replicate the bone microenvironment and its three-dimensional cell-laden matrix. Research in this area, thus far, has been restricted to models that lack these characteristics. Therefore, we argue that, while all the potential ramifications of the proposed approach are not all characterized in this paper, there are extensive applications to this model that could influence thinking beyond the scope of the specialized community.

3) The findings and results are original, but the conclusions are overinflated and not substantiated by the results provided for the following reasons: A serious concern is that the conclusions are all based on very small sample sizes (N=4 for FTIR, ROS and N=3 for SEM, Live-Dead and PCR. No sample sizes mentioned for TEM and serial block SEM). It is not clear whether the experiments were repeated, or these were all done in the same timeline. This not provide confidence in the scientific rigour and repeatability of these observations.

Response: We acknowledge the reviewer's concern and have attempted to better substantiate our claims and their potential limitations in this revision, in an effort to eliminate overinflated claims.

We also acknowledge the reviewer's concern regarding scientific rigor and repeatability. We have provided additional details regarding sample size in each section of the paper, and have included additional details to the sections where sample size was not originally reported. We regret this unintentional oversight in the original submission.

To ensure greater repeatability and rigor, we have increased the sample size of our experiments and have added supporting data using complementary methods to ensure that the effects seen are reproducible across multiple analyses methods. We have also redistributed a significant component of our data in an effort to increase the clarity of the time-scales presented in each experiment, and the specific variables being tested. For instance, in figure 1, we increased the sample size of the FTIR data, the quantification of mineral:matrix ratio, as well crystallinity index to include at least 6 replicates. We have also generated new data for crystallographic analyses via TEM/Selective Area Electron Diffraction, SEM, and EDX. These are now included either in figure 1 or in supplementary figures S3-S6. Additionally, we have doubled the sample size for experiments quantifying ROS, live and dead, and cell proliferation, and added results for OIM-treated samples. These are now shown in a separate figure in the main paper (Figure 2). Regarding the analyses of cell function reported in figure 4 of the original paper, we have now complemented the data to include additional cell surface markers, the quantification of protein expression, and percentage of expressing cells in each group (Figure 3). We also reported images of multiple locations per samples in our reflectance microscopy data to illustrate reproducibility and consistency of the results regarding the cell dendrite effect that we observed, and included data for quantification and imaging of dendrite-like structures in OIM-treated samples. These are included in supplementary figure S15. Moreover, we performed alizarin red analyses for at least 6 more replicates per group for 3 time points, added intracellular calcium data for days 7 and 14 in both mineralized and OIM treated samples. We have also included additional block face serial SEM slice images of non-mineralized, mineralized and OIM-treated samples, which is now presented after the results of cell function in the main paper (Figure 4), as well as in the supplementary material (Supplementary Figure 16). In addition, we have now included in our supplementary results qualitative data from real-time cell morphology analyses comparing cells embedded in mineralized hydrogels and positive controls after 7 days of culture, have determined cell proliferation via an Alamar Blue assay up to 21 days, have included quantification of collagen diameter, demonstrated controllable start and stop of matrix calcification via alizarin red staining, provided qualitative confocal fluorescence microscopy of both intra and extra-cellular mineral, determined the nanoscale mechanical properties of the mineralized samples in wet and dry states via AFM, as well as macroscale mechanical characterization of our constructs via a rheometer, and have characterized the expression of cell-surface DMP1 in pre-osteocyte differentiated hMSCs in vascularized samples, per the reviewer's request.

Finally, we point out that this study accumulates over two years of diligent experimental optimization, and follows extensive repetition of sample preparation and data collection, in addition to what was ultimately included in this report. In total, we provide data from the analyses of nearly 200 individual mineralized samples, which demonstrates the highly reproducible character of the proposed strategy. Also, we contend that despite the fact that a small set of experiments had a sample size of limited number, the effects observed in these experiments were confirmed and validated across multiple different methods. For instance, cell differentiation was assessed via qPCR, multiplex ELISA array, immunohistochemistry, quantification of percentage of protein expression, as well as expression intensity for a total of 12 proteins and 4 genes. All methods confirmed the same effect - that matrix mineralization results in comparable levels of differentiation as OIM controls. In another example, mineralization was characterized using 11 different methods: 1. FTIR (Spectra, mineral:matrix ratio, and crystallinity index), 2. EDX, 3. Standard SEM, 4. Serial Block-face backscattered SEM (in 3D and 2D), 5. TEM, 6. SAED, 8. Fluorescent Intracellular and 7. Extracellular calcium, 9. Alizarin red, 10. Von Kossa staining and 11. AFM/nanoindentation. All methods confirmed the ability of the proposed strategy to mineralize the cell-laden collagen hydrogels. Furthermore, all protocols followed well-accepted methods of statistical analysis, and therefore, we are confident of the validity and reproducibility of our data.

5) There were no positive osteogenic induction media controls used for any of the in vitro studies of nanostructure, composition and cytotoxicity, morphology or osteocyte-like features (see Figure 1-3). This is important to substantiate how this compares to gold standard approaches, which poorly induce mineralisation when compared to the levels and organisation in vivo.

Response: The main reason the nanostructure and compositional characterization experiments in Figures 1 did not originally include OIM as a control is because those experiments were designed to specifically assess the ability of the calcium and phosphate/mOPN-supplemented medium to mineralize the collagen matrix independent of mineral secretion by cells – hence the 3-day time point of the analyses. The hypothesis being tested was not whether the matrix mineralization (which happens in 3 days prior to calcium-secretion by cells) would compare to that obtained by OIM (which takes at least 14 days), but rather whether the mineralization was comparable in nanostructure and composition to that of native bone. This was one of the original objectives of the study. Thus, from a study design stand-point, we reasoned that the most adequate positive control for Figures 1 was the native bone itself, which was used for that section of the paper. Having said that, to ensure a more complete analysis of our data, as suggested by the reviewer, we have now included TEM images and selective area electron diffraction (SAED) data of OIM-treated cells (Supplementary Figure S4), as well as new SEM/EDX data for OIM-treated samples (Supplementary Figure S3). To account for the comparison of OIM versus mineralized samples regarding cytotoxicity, we have included data for live/dead (Figure 2a-d), ROS (Figure 2b), and Alamar blue (Figure 2f) for the OIM control group. Concerning the characterization of osteocyte-like (dendrite) features, we have added data for OIM-treated samples in our cell morphology experiments, and have combined those figures with other morphological analyses of immunostaining in (Fig. 3n, q). We have also collected serial block face backscattered SEM images of OIM-treated samples (Figure 4c, and Supplementary Figure S16), although 3D reconstructions were only performed for the mineralized group due to the highly labor-intensive nature of the method (~1 wk processing and 24-48 hr of continuous microscope time per sample), and the high-cost associated with sample data collection and 3D reconstruction (>\$3000 per 3D image). In addition, those images characterize 3D mineral formation, which is well established to be absent after 7 days of culture in OIM or non-mineralized collagen controls. Figure 1 has been modified accordingly:

Figure 1: Nanostructure, composition and characterization of mineralized collagen. SEM images of (a) non-mineralized and (b) mineralized collagen showing the formation of fibril bundles. Extrafibrillar mineral is apparent in (b). (N=4) Scale bar: 400 nm. (Insets) Collagen hydrogels prior to (a) and immediately after (b) a 3-day mineralization period. The mineral formation resulted in a white opaque appearance. c) EDX spectra of mineralized samples confirmed the presence of Ca and P in mineralized specimens (bottom), and lack thereof in non-mineralized controls (top). (N=4) TEM images of (d) non-mineralized collagen showed the typical D-banding pattern of collagen, while the banding becomes obscured in the (e) mineralized collagen, which appears to have both intra and extrafibrillar mineral crystallites (Supplementary movie 1). Scale bar: 500 nm. (N=4) (f) The zoomed view of a single mineralized fibril suggests the preferential orientation of intrafibrillar mineral apatite crystallites (dark streaks) in the (001) position, parallel to the c-axis of the fibril. Scale bar: 50 nm. (g) Selective Area Electron Diffraction (SAED) of a mineralized collagen shows the typical broad arcs for the (002) plane and overlapping arcs for the (112), (211), and (300) planes that are consistent with the known hexagonal crystalline structure of hydroxyapatite. (h) FTIR spectra and respective (i) mineral:matrix ratio (**** $p < 0.0001$, ANOVA/Tukey) and (j) crystallinity index (** $p < 0.001$, ANOVA/Tukey) of mineralized and non-mineralized collagen constructs, dotted lines are a reference values of native bone. (N=6) (k) AFM nanoindentation modulus of non-mineralized and mineralized hydrated collagen fibrils.

*Mineralization resulted in over a 1000-fold increase in stiffness for individual fibrils (**** $p > 0.0001$, Student's *t*-test) ($N=3$). Dotted line is a reference value of native mineralized collagen.³⁸*

6) Based on the gene expression studies later, while there are increases for some genes with the mineralization media (versus OIM), these do not concur with the timelines of FTIR that suggests it is identical to real bone at 7 days (whereas gene expression is lower than OIM at 7 days – which cannot recapitulate real bone).

Response: This is correct. That is because the FTIR data compares mineralized bone against our collagen mineralized using medium-induced calcification process. Therefore, matrix calcification, being induced and controlled by the medium, precedes upregulation of osteogenic genes, and ultimately results in cell differentiation in the absence of any other supplement.

7) The authors make major claims regarding the extent of mineralisation, which hang largely on the image provided of Alizarin Red staining (Figure 4c) and very limited FTIR/EDX data and immunostaining.

Response: In addition to Alizarin red and FTIR/EDX data, we have provided SEM data for mineralized samples acquired via surface analyses (Supplementary Figure S3), focused-ion beam milled collagen fibrils (Supplementary Figure S16), as well as blockface backscattered views of pericellular mineralization (Figure 4 and Supplementary Figure S16). We showed TEM images of pulverized mineralized samples (Supplementary Figure S5), and TEM characterization of gradually mineralizing samples from day 1 – day 3 (Supplementary Figure S6 and Supplementary Movie S1). We have also reported on the 3D reconstruction and quantification of mineralized collagen fibers as determined by the backscattered density of block-face microtomed SEM images (Figure 4), which to the best of our knowledge, is the first time that is reported. This follows an extensive process of segmentation of 190 individual 60-nm sections followed by 3D reconstruction of the mineralized matrix. Moreover, we have now included Selective Area Electron Diffraction (SAED) data to determine the crystallinity of individual apatite crystals within the mineralized fibers and respective controls (Fig 1g and Supplementary Figure S4), and have provided qualitative and quantitative data of the formation of intracellular and extracellular calcium via fluorescent microscopy (Fig. S18). In summary, mineralization was characterized using 11 different methods: 1. FTIR (Spectra, mineral:matrix ratio, and crystallinity index), 2. EDX, 3. Standard SEM, 4. Serial Block-face backscattered SEM (in 3D and 2D), 5. TEM, 6. SAED, 8. Fluorescent Intracellular and 7. Extracellular calcium, 9. Alizarin red, 10. Von Kossa staining and 11. AFM/nanoindentation, which represents quite a robust and consistent set of mineral characterization data.

8) The biggest concern is that Figure 4c is an extremely poor quality image (the image appears to be out of focus and/or poorly processed). Is it possible that the red hue is due to the presence of the CaP in the media, rather than deposition on the matrix?

Response: We regret that Figure 4c was processed to appear with poor quality. To certify the appearance and reproducibility of mineralized samples as characterized by Alizarin Red, a set of Alizarin Red images from days 1-21 are provided below for a more detailed assessment. We have also replaced the image in the figure 4c, which is now reported in the supplementary data (Supplementary Figure S11).

Supplementary Figure S11:

Supplementary Figure S11: Representative alizarin red staining images depicting homogenous calcification in the mineralized constructs within 7 days in culture. A total of 5×10^4 cells were encapsulated in collagen (1.5 mg/ml) hydrogels (N=4). For the CaP condition, cells were cultured in DMEM supplemented with 4.5 mM of CaCl_2 and 2.1 mM of K_2HPO_4 . For the mOPN condition, DMEM supplemented with 100 $\mu\text{g}/\text{mL}$ of mOPN was used; In OIM group, the cells were cultured in DMEM containing 100 nM Dexamethasone, 50 μM ascorbic acid and 10 mM β -glycerol phosphate. Non-mineralized constructs were cultured in non-supplemented complete DMEM

medium. After 1, 7 and 21 of culture, Alizarin Red S (Sigma) staining was performed to assess the overall mineralization. In brief, after culture, samples were washed in PBS, fixed in 4% paraformaldehyde for 5 min and then stained using 2% (w/v) Alizarin red solution at pH 4.2 for 5 minutes. After repeated washing in distilled water to remove any unbound stain, the constructs were imaged in bright field mode. Non-mineralized constructs showed no signs of mineral deposition even after 21 days, whereas the constructs exposed to osteoinductive supplements showed mineral nodule formation after 21 days of culture. The constructs treated with CaP containing medium resulted in the random deposition of mineral nodules, while treatment with mOPN alone did not induce any visible mineral deposition. Of note, intense red staining was first detected in the mineralized construct at day 3 (Supplementary Figure S2), maintained at day 7 and day 21, suggesting the presence of dense calcium phosphate deposits uniformly distributed throughout the matrix. Scale bar: 100 μ m

9) The staining is also not consistent with the gene expression, which shows OCN levels to be higher at day 7 in the osteogenic control (see Figure 4a).

Response: Gene expression of OCN is significantly higher for OIM versus non-mineralized controls on day 7, but not versus the mineralized group. It increases to comparable levels on day 14, and demonstrates a marked increase for the mineralized group on day 21. The immunostaining image in the paper (reproduced below) shows representative data for day 14 (not day 7), where both OIM and mineralized samples had statistically comparable gene expression for OCN. Additional immunostaining images of days 7 and 21 are shown in supplementary information (Supplementary Figure S12). We have now added quantification of protein expression via both fluorescence intensity and well as % of OCN⁺ cells. The following data was added to ensure a more robust characterization:

Excerpt of OCN gene expression and protein characterization via immunostaining, as depicted in the main paper.

Supplementary Figure S12: Expression of osteocalcin in hMSCs encapsulated within non-mineralized versus mineralized hydrogels, as detected by immunofluorescence staining (a). Cells treated with osteoinductive medium served as the positive control. In non-mineralized constructs, OCN expression was minimally detected at the early time point. However, prolonged culture for 21 days within the construct resulted in a slight increase in the OCN expression level. OIM and mineralized constructs, on the other hand readily exhibited intense OCN expression, about 6-fold higher than those in non-mineralized constructs (b), within 7 days of culture and almost 80% differentiation by 14 days (c). Unlike mineralized constructs, that retained OCN expression level after 21 days, cells in OIM had a markedly reduced expression. (* $p < 0.05$; ** $p < 0.01$, * $p < 0.001$; **** $p < 0.0001$, by ANOVA test; Data represented as mean \pm SD; (N=3); Scale bar: 100 μ m).

10) Moreover, the OCN immunostaining provided in Figure 4(d) is over-processed and shows no difference between mineralised and osteogenic positive control at day 14.

Response: We have re-processed the confocal images in the study to ensure that there is no over-processing. The image has now been replaced with what currently is Figure 3c-e, as depicted below.

11) The methods for the EDX studies are not described.

Response: The methodology for EDX analysis has been included now in the revised manuscript. The following was added: “For SEM analysis, samples were fixed with 2.5 % glutaraldehyde for 1 h at room temperature, washed in distilled water and subjected to a series of ethanol dehydration steps for 10 min each. Subsequently, the samples were critical point dried, sputter coated with gold/palladium and observed under SEM (FEI Helios Nanolab™ 660 DualBeam™) (N=6). The elemental analysis for the presence of Ca and P was carried out using the attached EDX detector (energy dispersive X-ray spectroscopy; INCA, Oxford Instruments) (N=4).”

12) Additional quantitative evidence of the increase in intracellular and extracellular mineralisation are required to support the conclusions of this paper (e.g. quantitative calcium assay, additional independent staining).

Response: We appreciate the reviewer’s recommendation and agree that the addition of intracellular and extracellular mineralization will strengthen the conclusions of the paper. To address this concern, dual calcium indicators (cell permeant Fluo-4AM dye (494/506 nm) and Rhod-5N dye (551/576 nm)) were used to distinguish intracellular and extracellular Ca²⁺, respectively. After 7 and 14 days of culture in mineralized and OIM treated collagen matrix, hMSCs were loaded with 5 μM of Fluo-4 AM for 2 hours. Post incubation, excess Fluo-4 AM was rinsed off, followed by loading of 5 μM of Rhod-5N and the samples were subsequently imaged live to visualize the increase in fluorescence intensity upon binding to Ca²⁺ using LSM 880 Laser scanning Confocal microscope.

As a first step to determine how our mineralization approach affects the intracellular and extracellular mineralization, we examined the intracellular levels of calcium in cells encapsulated in mineralized and OIM groups. A significant increase in cytosolic calcium was observed in mineralized groups, with over a two-fold increase in fluorescence expression compared to OIM treated cells. Concomitant live imaging of extracellular calcium deposition on day 7 revealed intense staining throughout the mineralized matrix. Conversely, no trace of extracellular calcium was noticed in the OIM treated collagen samples. Upon extending the culture to day 14, the intracellular calcium signal was stronger in OIM treated cells compared to mineralized samples. Nevertheless, after 14 days, random extracellular calcium deposits were identified in the vicinity of the cells treated with OIM, substantiating previous findings that the mineral precursors are synthesized and secreted by cells following the continued exposure to osteoinductive supplements. For mineralized samples, on the other hand, the matrix showed a relatively homogenous distribution of calcium deposition after 14 days. This data was added to the supplementary information together with a brief discussion in the main paper.

Supplementary Figure S18: Dual calcium indicators (cell permeant Fluo-4AM dye (494/506 nm) and Rhod-5N dye (551/576 nm)) were used to distinguish intracellular and extracellular Ca²⁺, respectively. After 7 and 14 days of culture in mineralized and OIM treated collagen matrix, hMSCs were loaded with 5 M of Fluo-4 AM for 2 hours. Post incubation, excess Fluo-4 AM was rinsed off, followed by loading of 5 μM of Rhod-5N and the samples were subsequently imaged live to visualize the increase in fluorescence intensity upon binding to Ca²⁺ using LSM 880 Laser scanning Confocal microscope (N=4). Cells cultured in mineralized collagen (a-c) had a two-fold increase in cytosolic calcium (green) on day 7, compared to those cultured in OIM (*p<0.05). For OIM, these levels were only attained on day 14 (j-l). Extracellular calcium (red) appears to be distributed throughout the matrix since day 7 in mineralized collagen (b-c, e-f), with higher intensity spots surrounding cells on day 14, possibly due to cell-secreted minerals. OIM samples had very little extracellular calcium on day 7 (h-i) and only localized calcium deposits in the pericellular regions on day 14 (k-l). Scale bar: 50 μm

13) Further to this point, the authors claim that “Mineralized samples had comparable values to that of native bone” after 7 days. This is simply not possible in terms of time required for the cells to make that mineral, so it can only be assumed that this is deposited from the media, but this has not been clearly articulated in the paper. Moreover, they make this claim on the basis of the mineral:matrix ratio, rather than any quantitative comparison of the total mineral in their construct versus real bone (you could have very little mineral and very little matrix and the mineral:matrix ratio would be similar). The authors have not adequately addressed or provided evidence whether the CaP have merely deposited on the scaffold from the media, or whether it enticed the cells to make a mineralized matrix themselves.

Response: We regret that the medium induced mineralization process, which is the key feature of our model, was not clearly articulated in the original version of the manuscript, despite our best efforts. To better present this process, we added a more descriptive sentence in the abstract and first mention of medium induced mineralization in the paper. These were further detailed in our answer to question 1. This, we hope, will emphasize the method of mineralization that we propose for the remainder of the paper.

With regards to the similarities to native bone, we reworded the sentence to read:

Page 7, line 7: *Similarly, although the absolute levels of mineral and collagen are inferior to those in native bone, given the low density of the collagen hydrogels in comparison to that of the native tissue, the relative mineral-to-matrix ratio (Figure 1i) and crystallinity index (Figure 1j) of the mineralized scaffolds approximated those of human bone.*

14) The authors claim that these constructs “mimic the nanaoscale structure, composition and function of the cell-rich and calcified bone matrix”. However, in the absence of any mechanical testing data, it is not possible to support the claim that the constructs can recapitulate the load bearing capacity of bone, which is the primary function of bone and is also critical to the biological behaviour of the cells within the construct. This is a serious limitation. For this tissue construct to be a functional representation of bone, they need to provide evidence that the mineral observed is bound to the matrix, and that there is sufficient mineralisation to enhance the mechanical properties. Indeed, a current limitation of most bone-like models is their failure to induce sufficient mineral or recapitulate the mechanical properties of bone. There is insufficient evidence that the study reported here has overcome these limitations.

Response: We fully agree with the reviewer, and have modified our statements to temper the claims of mimicked bone function. We have now exclusively referred to bone functions that are directly represented by our data, which are largely linked to upregulation of osteogenic markers, the secretion of soluble factors, and paracrine homing of prostate cancer cells. To temper these claims of mimicked function and better reflect the function that we characterize, we have also reworded the title of the paper to read: “*Rapid fabrication of vascularized cell-laden bone models with biomimetic intrafibrillar collagen mineralization, nanostructure and composition*”.

In addition to these observations, we contend that there are two levels of mechanobiology that are important to be taken into account with regard to mechanical regulation of bone function. On a tissue macroscale, load-bearing function is determined by the overall composition, density and mineralization of the tissue. Due to the low density of the collagen hydrogels that we utilize (1.5 mg/mL), or virtually any other low-density gel system, it is virtually impossible to achieve the GPa-range of stiffness that native bone has. This, as the reviewer pointed out, greatly limits the ability of our model to replicate the load bearing function of native bone, and is not a claim that we make with our strategy. We have reworded our statements throughout the paper to reflect this interpretation of our results.

Secondly, on a more specific scale of mechanotransduction, it is known that cells respond to localized microscale stiffness more so than the overall stiffness of an entire tissue [Mechanosensing by the nucleus: From pathways to scaling relationships. Cho, Irianto, and Discher. J. Cell Biol. Vol. 216 No. 2 305–315]. Therefore, we sought to characterize the increase in stiffness of individual collagen fibrils before and after mineralization to determine whether, at the local level, the addition of apatite crystals to the collagen fibers would approximate the range of stiffness of native bone collagen. AFM nanoindentation data shows that while demineralized collagen had a stiffness of about 0.0016 ± 0.0003 GPa, mineralized collagen reached a stiffness range on the order of 2.21 ± 0.046 GPa, which is about 1000X stiffer than non-mineralized controls.

This is in support of our nanostructural characterization that shows mineral formation both within and on top of collagen fibrils, thus illustrating the direct binding of the apatite crystals to the collagen fibrils.

We have added the following to the paper:

Page 7, line 11: *To determine the ability of the apatite mineral to bind to and mechanically reinforce the fibrils, we performed AFM nanoindentation on individual collagen fibrils in solution and ambient air (Supplementary Figure S8), either before or after mineralization. The hydrated non-mineralized collagen had an elastic modulus of 0.0016 ± 0.0003 GPa, while the elastic modulus of mineralized fibrils was 2.21 ± 0.046 GPa (Figure 1k). Those values are consistent with previous reports of nanoindentation modulus of non-mineralized collagen,³⁷ and collagen extracted from tooth dentin,³⁸ which has a nearly identical mineralization profile to native bone, and also closely approximate the values reported for mineralized bone collagen measured using AFM force spectroscopy (pulling).³⁹ Despite the significant increase in nanoindentation modulus of individual fibrils, the overall stiffness of the hydrogel constructs was still markedly lower than that of bone, which is in the order of 20 GPa on a macroscale (Supplementary Figure S8).*

*Supplementary Figure S8: Analysis of nanoindentation elastic modulus of non-mineralized and mineralized collagen fibrils performed in water (a-c) and in air (d-f) using an atomic force microscope (Nanoscope 8 atomic force microscope, J scanner, Bruker). Representative load-displacement curves on (a) non-mineralized and (b) mineralized collagen fibrils, measured in water. The loading curve is represented in red and the unloading curve is represented in blue. The dotted line corresponds to the indentation on fibrils, while the indentation on the adjacent mica substrate is denoted by the solid line. Inset shows the representative AFM topographic image of individual fibrils fixed onto freshly cleaved muscovite mica disc, with the color scale on the right corresponding to the height. (c) Comparison of average elastic modulus in non-mineralized versus mineralized fibrils, calculated by fitting the Hertz model to the loading curves. An increase of over 1000-fold was recorded in hydrated mineralized fibrils versus non-mineralized controls (**** $p < 0.0001$, Student's *t*-test). Corresponding load-displacement curves and average modulus of non-mineralized and mineralized fibrils in air are shown in (d), (e) and (f), respectively. (f) The nanoscale elastic moduli values in air followed the same trend, with a substantial increase*

*in stiffness in mineralized fibrils compared to its non-mineralized counterparts (**** $p < 0.0001$, Student's t -test). In all cases, the maximum penetration depth was set to ~ 15% of the fibril thickness. In (b-e) the measurements were made using aluminum-coated, silicon microsphere tip of radius, 10 nm, resonance frequency of 300 kHz and a nominal spring constant of 26 N/m whereas for non-mineralized collagen fibril in (a), indentation was performed using triangular gold-coated silicon nitride tip of 30 nm radius, 65 kHz resonance frequency and 0.35 N/m nominal spring constant. Measurements were performed on 5 to 10 different locations from at least 3 fibrils per sample, with a total of 3 samples per condition. (g) The bulk elasticity (shear storage moduli) of mineralized hydrogels was significantly higher compared to their non-mineralized controls (* $p > 0.05$, Student's t -test). An oscillatory rheometer fitted with an 8 mm parallel plate geometry was used to measure the storage modulus and loss modulus in frequency sweep mode at 1 Hz frequency and 1% strain (N=4).*

15) The logic and timelines investigated are not clear throughout. Some of the parameters were only investigated for 7 days, and it in other cases it is not clear from which precise timepoint data is being presented (e.g. Figure 2 and 3).

Response: The logic for the timelines presented was the following: We first sought to characterize the nanostructural and compositional characteristics of the mineralized samples after mineralization was complete (3 days). Since these pertained to the medium-induced mineralization only, and not to cell-dependent processes, samples were characterized immediately after 3 days. To increase clarity of these timelines, we have separated the nanostructural and mineral characterization that are conducted on 3-day mineralized samples in figure 1, from other cell studies of cytocompatibility that are conducted for up to 21 days. The latter are now presented in figure 2. After determining that the process that led to successful mineralization is not cytotoxic (Figure 2), we then sought to characterize whether differentiation was comparable to a known standard, where OIM was utilized. Those particular experiments were conducted for up to 21 days, in order to characterize osteogenic cell differentiation and compare to standard osteogenic protocols, which are typically carried out for that duration of time.

16) The authors claim that the “pericellular empty space....is reminiscent of lacunae in osteonal bone”. This is not correct, Fig. 3 depicts pericellular empty space, which do not occur in vivo; it has been widely published that the PCM of osteocytes fills the pericellular space (studies by Schaffler, Weinbaum, You, McNamara). It is likely that these empty spaces have arisen due to poor fixation of the tissue.

Response: We appreciate the reviewer's comments and insights into these findings, which we were not aware of. We have now removed the statements that associated the block face SEM images being reminiscent of lacunae in native bone, and refer to these structures as likely artifacts. The figures and captions now read:

Figure 4: 3D volumetric reconstruction of backscattered electron micrographs obtained via serial block-face SEM. a) Matrix surrounding cells in non-mineralized collagen had little backscattered contrast, suggestive of lack of mineralization. b) In mineralized hydrogels the matrix was visibly darker due to the backscattered electron contrast of mineralized fibrils, especially in the matrix immediately surrounding the cells. c) Collagen in OIM-treated samples also lacked significant backscattered electron signal. d) Illustration of the serial stacking of 190 60 nm-thin sections, the segmentation of cells (blue) from the surrounding mineralized matrix (middle panel, scale bar: 20 μm), and visualization of block 3D image (right panel). Arrows in (d) show narrow dendrite-like cell processes. e) 3D rendered image showing cells (blue) embedded in mineral (red), with the underlying collagen (grey). f) Exclusion of collagen via digital processing illustrates the density of mineralized collagen, and cells spread within a bed of mineralized matrix. Narrow cell processes (arrows) shown in higher magnification in (g) appear to extend between mineralized fibrils (Supplementary Video 2) (scale bar: 10 μm). e) Digital removal of cell bodies from within the mineralized matrix illustrates density of mineral surrounding the cell structures. The total length of the x-axis in all 3D reconstructions (d, e, f and h) is 62 μm.

Supplementary Figure S16: a-f) Backscattered SEM images acquired in serial block face imaging acquisition mode, from separate samples in non-mineralized, mineralized and OIM-treated groups. The dark contrast areas are due to the backscattered electrons emitted by minerals or denser features (i.e. cells and organelles), whereas the lower contrast areas are non-mineralized collagen or void spaces. Non-mineralized samples lacked any visible contrast (a-b). Mineralized samples had a markedly darker contrast throughout the matrix, and particularly greater mineral formation in the regions bordering the pericellular space, which is consistent the lamina limitans formed around osteocytes in osteonal bone (c-d). Matrix in OIM-treated samples had no visible backscattered contrast (e-f), consistent with lack of mineralization after 7 days in OIM medium. Void spaces around cells in (c-d) are likely artifacts from sectioning/processing. (g-h) Selected samples from mineralized group were also processed via focused ion beam (FIB) milling and imaged in higher magnification in backscattered mode to illustrate the density of the mineral contracts at the individual fibril level. Scale bar: 500 nm. (N=3) i) Collagen and mineral volume calculated from the 3D reconstructed image of cell-laden mineralized matrix showed that approximately 47% of the collagenous matrix was covered by mineral. For that, serial slices of resin embedded mineralized constructs obtained at 60 nm intervals using serial block face SEM was aligned and processed using AMIRA (Version 5) image analysis and reconstruction software. Subsequently, semi-automated segmentation of mineral and collagen was performed slice-by-slice, followed by volume rendering to compute the 3D volume occupied by the specific features of interest.

17) In general, the images provided do not provide any confidence that the cells have differentiated in widespread fashion (how many of the cells had dendrites?).

Response: To assess this, we have quantified the percentage of cells expressing PDPN, OCN and DMP1 on days 7, 14 and 21, which is shown in the graph below. The percentage of cells that expressed OCN in mineralized constructs, ranged from around 75-80%. The percentage expression of PDPN increased from approximately 40% on day 7 to approximately 75% on day 21 for mineralized samples. In Supplementary Figure S15 we also provide examples of 5

images from 4 different biological replicates from the mineralized group below to illustrate the widespread and consistent expression of cell dendrites over multiple samples/images.

Supplementary Figure S15: Quantification of the number (a) and length (b) of dendrite-like protrusions in hMSCs after 7 days of culture in 3D matrices. The cells were fixed, stained for F-actin and imaged using confocal microscopy (63x objective, Zeiss Airyscan LSM 880). The mean number of protrusions per cell was counted manually using the cell counter plugin in imageJ software. The average protrusion length was assessed by manually tracing the extensions with freehand selection tool and their lengths were measured with ImageJ. Protrusions that are less than 5 μm were excluded from the analysis. Cells in the mineralized group exhibited a significant increase in the number and length of protrusions compared to non-mineralized and OIM groups.

Data represented as Mean±SD, from four independent experiments with at least 35 cells per group. A set of images is shown to illustrate the consistency of the effect across samples and regions on a sample. Comparison by one-way Anova (***) $p < 0.001$). (c) Representative confocal images of cells from 4 different biological replicates exhibiting dendrite-like protrusions in 3D mineralized scaffolds after 7 days.

18) Figure 2 and the Supplementary Video 2 appears to depict a very tubular and very thick dendrite, which is not **reminiscent of an osteocyte**. Moreover, **no scale is given to allow independent assessment**.

Response: We agree with the reviewer that the long tubular extension in the video is not reminiscent of a dendrite. However, we point the reviewer's attention to the shorter and much smaller protrusions that appear later in the video and are now highlighted with arrows in figure 4g, which we argue are more representative of the osteocyte-like protrusions to which we referred. These are also depicted below. The scale bars for 2D images was added and the total X dimensions for the 3D projections were also added to the caption.

Figure 4: 3D volumetric reconstruction of backscattered electron micrographs obtained via serial block-face SEM. a) Matrix surrounding cells in non-mineralized collagen had little backscattered contrast, suggestive of lack of mineralization. b) In mineralized hydrogels the matrix was visibly darker due to the

backscattered electron contrast of mineralized fibrils, especially in the matrix immediately surrounding the cells. c) Collagen in OIM-treated samples also lacked significant backscattered electron signal. d) Illustration of the serial stacking of 190 60 nm-thin sections, the segmentation of cells (blue) from the surrounding mineralized matrix (middle panel, scale bar: 20 μm), and visualization of block 3D image (right panel). Arrows in (d) show narrow dendrite-like cell processes. e) 3D rendered image showing cells (blue) embedded in mineral (red), with the underlying collagen (grey). f) Exclusion of collagen via digital processing illustrates the density of mineralized collagen, and cells spread within a bed of mineralized matrix. Narrow cell processes (arrows) shown in higher magnification in (g) appear to extend between mineralized fibrils (Supplementary Video 2) (scale bar: 10 μm). e) Digital removal of cell bodies from within the mineralized matrix illustrates density of mineral surrounding the cell structures. The total length of the x-axis in all 3D reconstructions (d, e, f and h) is 62 μm .

19) Fig. 5 – there is no evidence provided to confirm whether the MSCs formed osteocytes when they were simultaneously cultured with HUVECS/neuroblastoma cells. DMP1/Sost staining could confirm that there was a bone cell network present simultaneously, but the authors only provided

Response: As per the reviewers' suggestion, we performed immunohistochemical evaluation for DMP1 expression in pre-vascularized and mineralized constructs after 7 days of culture in-vitro followed by 7 days implantation in-vivo. Non-mineralized samples showed low levels of DMP1 expression, consistent with gene and protein fluorescence analyses. Vasculature in both samples was visible by the presence of luminal structures that are consistent with the appearance of vasculature in histological sections (arrowheads). In the mineralized matrix, cells had more visible expression of DMP1 (arrows) and are seen near luminal structures.

Supplementary Figure S22: a) Engraftment of non-mineralized and (b) mineralized hydrogels in-vivo. H&E image showing erythrocyte filled capillaries (yellow arrow head) within the constructs after 3 weeks of implantation in nude mice. Scale bar: 500 μm . c) Expression of DMP1 was visible

in (c) non-mineralized and (d) mineralized constructs implanted with pre-formed vasculature after 7 days of culture in-vitro, followed by 7 days in-vivo. DMP1-expressing cells (yellow arrows), which appear more faint in non-mineralized constructs and more intense in mineralized hydrogels, are seen near cross-sectioned luminal structures that are consistent with the appearance of vasculature capillaries (yellow arrowheads), which are not stained. Scale bar: 100 μ m (N=4).

20) The term “on-demand” is overused and misleading. Moreover, there is no evidence of the process being “time-controllable, with the versatility of the synthesis being initiated and stopped on demand” – where is evidence of stopping the mineralisation process by the cells?

Response: To illustrate that our approach is time controllable, we have now included qualitative alizarin red staining images showing how the mineralization process can be initiated, terminated or resumed as required. By taking advantage of the versatility of our approach, we were able to trigger mineralization in constructs incorporating vascular and neural cells several days after encapsulating the cells.

Supplementary Fig. S19: Alizarin red staining showing the time-controlled mineralization of collagen hydrogel. a) Sample mineralization for 3 continuous days. Samples were then maintained in DMEM medium for the remainder of the experiment. b) Partial calcification of the collagen matrices was induced by exposing the samples to mineralizing medium (cell-culture medium with supersaturated Ca and P ions stabilized with mOPN) for 2 days. Next, the medium was replaced with standard DMEM without additional Ca and P or mOPN, to stop mineralization until day 4. Note the faint red staining till day 5 due to limited mineralization of the matrix. On day 5, samples were cultured with the Ca, P and mOPN rich medium again, thus resuming the mineralization process and completing matrix calcification, as visualized from the intense red staining at later time-points. After each time-point, the samples were fixed, stained with 2% Alizarin red S and imaged under bright field mode using EVOS FL Auto 2 Imaging System (N=6). Scale bar: 400 μ m.

21) The level of detail provided is inadequate, in particular regarding the timepoints depicted and scale bars for some of the figures.

Response: Each figure in the paper has been revised for timepoint description and to include scale bars.

22) The results provided are not convincing. Further evidence to strengthen the conclusions would require (1) an osteogenic positive control for comparison to the results presented in Figures 1-3, (2) quantitative evidence of the increase in intracellular and extracellular mineralisation are required (e.g. quantitative calcium assay, additional independent staining), (3) mechanical test data to confirm that the mineral deposited was mechanically integrated with the collagen matrix and provided a tissue close to that of bone, (4) evidence of the existence of a functional osteocyte network at the same time as the vascular/neuronal networks are established (DMP1/Sost staining).

Response: In addition to an extensive list of modifications and newly added data, we have addressed all 4 concerns listed by the reviewer: (1) We have included data from OIM controls for all the cell-based experiments in the original figures 1-3, (2) have provided quantitative evidence for the increase in intracellular calcium in mineralized samples after 7 and 14 days in culture and in comparison to OIM controls, (3) have provided mechanical characterization at macro and nanoscale via rheological tests and AFM nanoindentation, and (4) have provided evidence of DMP1⁺ cells adjacent to vascular structures in mineralized constructs.

23) The paper could influence thinking in the field if it were substantially moderated to remove the exaggerated conclusions (for e.g. “unprecedented levels of biomimicry”), in terms of functional mechanical and cellular behaviour and how this compares to real bone tissue.

Response: Thank you, and we have now significantly tempered our initial statements related to the above-mentioned claims (e.g the statement above now read: *Ultimately this approach allows for fabrication of bone-like tissue models with high levels of biomimicry that may have broad implications for disease modeling, drug discovery, and regenerative engineering.*)

Reviewers' comments:

Reviewer #1 (Remarks to the Author):

The authors have conducted a significant amount of new experiments that support previously unsupported claims and further enhance the quality and validity of the study.

A couple of minor points to include:

The goal of the study is an ambitious one and as I mentioned have provided new data to enhance the biological reliability of the model and the rigorosity of its characterization. However, given that there remain aspects that are not recreated as they are in vivo, it is important that the authors briefly mention and discuss aspects that remain have not been recreated in comparison to the in vivo scenario (for example, the chemical and physical composition of the organic matrix or the time of mineralization). This will not take away from the value of the study as it currently is, but instead will help elucidate the complexity of the challenge at hand while guide future investigations that hope to build on this important work.

Supplementary Figure S17 letters are capital, please put in same format as figure legend.

Page 17. The authors have attempted to identify matrix remodeling around the cells. Please give a reference to the statement that cell movement and protrusion is associated with matrix remodeling.

Reviewer #2 (Remarks to the Author):

The authors have made substantial revisions to the manuscript, which now provides convincing evidence in support of the effectiveness of the biomineralisation strategy to guide mineralization, vascularisation, innervation, and osteogenic differentiation. They have also revised the major claims of the article to temper the conclusions to more appropriately reflect the ability (and limitations) of the strategy to recreate bone mechanical function and in vivo osteocyte morphology. The authors have made a significant effort to clarify the novelty of their approach and elaborate on the relevance of their findings. The paper now represents a comprehensive and extensive body of work that will influence thinking in the field. The modifications and revisions of note, which have addressed this reviewers concerns, are outlined below:

- The authors have included data from osteogenic media controls for cell studies in figures 1-3
- The authors have provided quantitative evidence for the increase in intracellular calcium in mineralized samples after 7 and 14 days in culture and have compared these to osteogenic media controls
- The authors have determined the ability of the apatite mineral to mechanically reinforce the fibrils using AFM nanoindentation on individual collagen fibrils in solution and ambient air, and provide an appropriate reflection on the differences in the properties reported to that of native bone.
- The authors have included Selective Area Electron Diffraction (SAED) analyses of mineralized fibrils, which provides further evidence of the comparison to the crystal form of hydroxyapatite in native bone and in osteoblast-secreted apatite crystals
- The authors have quantified the percentage of cells expressing PDPN, OCN and DMP1 at specific time points (7, 14 and 21 days).
- The authors have performed additional immunohistochemical evaluation of DMP1 in pre-vascularized and mineralized constructs after 7 days in-vitro followed by 7 days in-vivo.
- The authors have included qualitative alizarin red staining images showing the timeline of mineralization and how this can be modulated by the presence/absence of medium with supersaturated Ca and P ions stabilized with mOPN.
- Each figure in the paper has been revised for timepoint description and to include scale bars.

- The authors have modified the paper title and specific statements to temper claims of mimicked bone mechanical function, and have clarified that their data confirms upregulation of osteogenic markers, secretion of soluble factors, and paracrine homing of prostate cancer cells.
- The authors have modified the paper to temper the original exaggerated conclusions by removal of the term “on demand” and revision of specific sentences, e.g. “Ultimately this approach allows for fabrication of bone-like tissue models with high levels of biomimicry that may have broad implications for disease modeling, drug discovery, and regenerative engineering”.

There is a remaining concern over the interpretation of the osteocyte morphology as is outlined below.

- Can the authors explain the different morphology in the Reflectance confocal microscopy images of F-Actin/DAPI stained hMSCs in mineralized (Figure 3m) cell-laden collagen? It appears that the cells have numerous very small filopodia projecting from the dendrites, which are not present in the non-mineralised and OIM samples.
- Further to this point the authors state that “a similar observation was noticed by Mata et al., who described greater filopodial extensions and osteoblastic differentiation in hole microtextures as opposed to groove microtextures, likely due to cells sensing these features as enclosed microenvironments with close cell-cell and cell-ECM contact”. Filipodia and dendrites are distinct cell structures that should not be assumed to be “similar”. Filipodia are not a distinct structure distinguish osteocytes. This comparison does not appear to be appropriate for this reason.
- More importantly, and related to both of the above points, the authors state that “non-mineralized (Figure 3l) or OIM-treated controls (Figure 3n), on the other hand, cells lacked any marked increase in the projection of cell dendrites (Supplementary Figure S15)”. However, the additional images provided in S15 appear to be only for the mineralised group, and the images in Figure 3(n) do not support the claim that these cells “lacked any marked increase in the projection of cell dendrites” or the quantitative data in S15. In Figure 3(m) no cell appears to have 27 +/- 10 dendrites (as indicated in the graph in S15 (a)), and thus it would appear that the authors must also be quantifying these very short filopodia-like protrusions. If this is indeed the case, this is incorrect, and the quantification needs to be redone to only quantify true dendrites and not filopodia. Further information on the approach used to identify dendrites to quantify the number (a) and length (b) of dendrite-like protrusions, and additional images for the non-mineralised and OIM cells are required to substantiate this aspect of the study.
- The reviewer does not agree that DMP1-expressing cells are more faint in non-mineralized hydrogels, nor that they are near the unstained vasculature capillaries. The differences in staining are very minimal between the two images, and many of the stained cells are far from the vascular-like channel. Some rephrasing here is warranted to temper the claims for this aspect of the study. Some additional comments that could enhance this manuscript are provided below:
 - Introduction: It would be appropriate to rephrase the sentence regarding bone targeting diseases by stating “We further validate this model system by showing that the engineered tissue can stimulate homing of engrafted prostate cancer cells in-vivo”
 - The authors provide in S15 five confocal image sets for actin, collagen and merged staining, should this caption have read “Representative confocal images of cells from 5 different biological replicates”, or are these five sample images from these 4 samples (i.e. 2 imaged regions from one replicate)?
 - It is this reviewers opinion that many of the important evidence in this paper is now contained in the supplementary figures (for example data S8 and S11, S12). It would be nice to see this data combined with figures in the main body of the paper.
 - For the images provided in Figure 4(e-f) it is requested to clarify that these are for mineralised conditions. Are such images available for the non-mineralised and OIM conditions and, if not, could this figure move to the Supplementary material to make way for the interesting data that is now in S8, S11 and S12?
 - S17: The figure heading reads “In (l) multiple narrow processes are shown in a cell (yellow arrow), and after 90 min, these processes have either disappeared or moved to another location (green arrows)”. Presumably the intent here was to refer to (M) but this has not been indicated in the text of the heading.

Reviewers' comments:

Reviewer #1 (Remarks to the Author):

The authors have conducted a significant amount of new experiments that support previously unsupported claims and further enhance the quality and validity of the study.

We thank the reviewer for his/her evaluation of our manuscript and acknowledge that his/her suggestions have improved the quality of our study. We have addressed the concerns outlined by the reviewer both in the manuscript and in the response in blue.

A couple of minor points to include:

1. The goal of the study is an ambitious one and as I mentioned have provided new data to enhance the biological reliability of the model and the rigorosity of its characterization. However, given that there remain aspects that are not recreated as they are in vivo, it is important that the authors briefly mention and discuss aspects that remain have not been recreated in comparison to the in vivo scenario (for example, the chemical and physical composition of the organic matrix or the time of mineralization). This will not take away from the value of the study as it currently is, but instead will help elucidate the complexity of the challenge at hand while guide future investigations that hope to build on this important work.

We agree with the reviewer's opinion in this regard and have included this discussion in our manuscript:

Page 7, Line 21: *Overall, both the mineral composition and nanostructural organization approximate that of a loosely packed, woven bone tissue. Nevertheless, there are still important compositional, structural and mechanical characteristics that would need to be optimized to fully mimic the expected function of mature human bone. These form the basis for future studies utilizing the process that we describe here.*

2. Supplementary Figure S17 letters are capital, please put in same format as figure legend.
We have double checked this suggestion, and the figure legend matches the captions.
3. Page 17. The authors have attempted to identify matrix remodeling around the cells. Please give a reference to the statement that cell movement and protrusion is associated with matrix remodeling.

We have modified the statement to read: Page 17, Line 6: *This indicates that even though approximately 50% of the organic matrix was mineralized (Supplementary Figure S16), cells were still able to move within the surrounding matrix (Supplementary Figure S17, Supplementary Videos 3, 4), secrete soluble proteins as well as process intracellular and extracellular calcium (Supplementary Figure S18), all of which are indicative of active new tissue formation.*

Reviewer #2 (Remarks to the Author):

The authors have made substantial revisions to the manuscript, which now provides convincing evidence in support of the effectiveness of the biomineralisation strategy to guide mineralization, vascularisation, innervation, and osteogenic differentiation. They have also revised the major claims of the article to temper the conclusions to more appropriately reflect the ability (and limitations) of the strategy to recreate bone mechanical function and in vivo osteocyte morphology. The authors have made a significant effort to clarify the novelty of their approach and elaborate on the relevance of their findings. The paper now represents a comprehensive and extensive body of work that will influence thinking in the field. The modifications and revisions of note, which have addressed this reviewers concerns, are outlined below:

- The authors have included data from osteogenic media controls for cell studies in figures 1-3
- The authors have provided quantitative evidence for the increase in intracellular calcium in mineralized samples after 7 and 14 days in culture and have compared these to osteogenic media controls
- The authors have determined the ability of the apatite mineral to mechanically reinforce the fibrils using AFM nanoindentation on individual collagen fibrils in solution and ambient air, and provide an appropriate reflection on the differences in the properties reported to that of native bone.
- The authors have included Selective Area Electron Diffraction (SAED) analyses of mineralized fibrils, which provides further evidence of the comparison to the crystal form of hydroxyapatite in native bone and in osteoblast-secreted apatite crystals

- The authors have quantified the percentage of cells expressing PDPN, OCN and DMP1 at specific time points (7, 14 and 21 days).
- The authors have performed additional immunohistochemical evaluation of DMP1 in pre-vascularized and mineralized constructs after 7 days in-vitro followed by 7 days in-vivo.
- The authors have included qualitative alizarin red staining images showing the timeline of mineralization and how this can be modulated by the presence/absence of medium with supersaturated Ca and P ions stabilized with mOPN.
- Each figure in the paper has been revised for timepoint description and to include scale bars.
- The authors have modified the paper title and specific statements to temper claims of mimicked bone mechanical function, and have clarified that their data confirms upregulation of osteogenic markers, secretion of soluble factors, and paracrine homing of prostate cancer cells.
- The authors have modified the paper to temper the original exaggerated conclusions by removal of the term “on demand” and revision of specific sentences, e.g. “Ultimately this approach allows for fabrication of bone-like tissue models with high levels of biomimicry that may have broad implications for disease modeling, drug discovery, and regenerative engineering”.

We appreciate the reviewer’s careful evaluation of our efforts and appreciate his/her valuable input towards improving our manuscript.

There is a remaining concern over the interpretation of the osteocyte morphology as is outlined below.

1. Can the authors explain the different morphology in the Reflectance confocal microscopy images of F-Actin/DAPI stained hMSCs in mineralized (Figure 3m) cell-laden collagen? It appears that the cells have numerous very small filopodia projecting from the dendrites, which are not present in the non-mineralised and OIM samples. The review on osteocytic dendritic processes by Bonewald (J Musculoskelet Neuronal Interact 2005; 5(4):321-324) suggests that dendrite formation in osteocytes is preceded by cellular processes that orient the formation of dendrites. Therefore, although we acknowledge that filopodia and dendrites are different structures, it is possible that the filopodial projections which are visibly present in greater excess in the cells in the mineralized conditions serve this purpose. This is discussed both in the main paper and in the caption to Supplementary Figure S15. Page 14, Line 1: *Consistent with that notion, fluorescence images obtained from cells cultured for at least 7 days in the mineralized microenvironment, showed cells extending narrow actin-rich dendritic-like processes that appeared to radiate through the matrix toward neighboring cells, and in some cases, with tiny extensions stemming from them (Figure 3p). These morphological features resemble the characteristic morphology of osteocytes, where cell processes precede and guide dendrite growth and orientation⁶⁰.*
Supplementary Figure S15: *Quantification of the number (a) and length (b) of dendrite-like protrusions in hMSCs after 7 days of culture in 3D matrices. The cells were fixed, stained for F-actin and imaged using confocal microscopy (63x objective, Zeiss 210 Airyscan LSM 880). The mean number of protrusions per cell was counted manually using the cell counter plugin in imageJ software. The average protrusion length was assessed by manually tracing the extensions with freehand selection tool and their lengths were measured with ImageJ. Thin filopodial structures of length and width less than 5 μm and 1 μm, respectively were excluded from the data analysis for dendrite-like protrusions. Cells in the mineralized group exhibited a significant increase in the number and length of protrusions compared to non-mineralized and OIM groups. Data represented as Mean±SD, from four independent experiments with at least 30 cells per group. Comparison by one-way Anova (**p<0.01, ***p<0.001). Representative confocal images of cells from 4 different biological replicates exhibiting dendrite-like protrusions in 3D non-mineralized (c), mineralized (d) and OIM (e) constructs after 7 days. Small extensions stemming from the protrusions can be particularly noted in cells encapsulated within the mineralized matrix which may play a role in orienting dendrite formation and growth⁶⁰.*
2. Further to this point the authors state that “a similar observation was noticed by Mata et al., who described greater filopodial extensions and osteoblastic differentiation in hole microtextures as opposed to groove microtextures, likely due to cells sensing these features as enclosed microenvironments with close cell-cell and cell-ECM contact”. Filopodia and dendrites are distinct cell structures that should not be assumed to be “similar”. Filopodia are not a distinct structure distinguish osteocytes. This comparison does not appear to be appropriate for this reason.
The reviewer raises a valid point and the terminology has been modified to adopt ‘processes’ instead of ‘filopodia’, as done by the authors in the referenced paper. The text now reads:
Page 14, Line 4: *A similar observation was noticed by Mata et al., who described more numerous cell processes and osteoblastic differentiation in hole microtextures as opposed to groove microtextures, likely due to cells sensing these features as enclosed microenvironments with close cell-cell and cell-ECM contact.*

3. More importantly, and related to both of the above points, the authors state that “non-mineralized (Figure 3l) or OIM-treated controls (Figure 3n), on the other hand, cells lacked any marked increase in the projection of cell dendrites (Supplementary Figure S15)”. However, the additional images provided in S15 appear to be only for the mineralised group, and the images in Figure 3(n) do not support the claim that these cells “lacked any marked increase in the projection of cell dendrites” or the quantitative data in S15. In Figure 3(m) no cell appears to have 27 +/- 10 dendrites (as indicated in the graph in S15 (a)), and thus it would appear that the authors must also be quantifying these very short filopodia-like protrusions. If this is indeed the case, this is incorrect, and the quantification needs to be redone to only quantify true dendrites and not filopodia. Further information on the approach used to identify dendrites to quantify the number (a) and length (b) of dendrite-like protrusions, and additional images for the non-mineralised and OIM cells are required to substantiate this aspect of the study. We appreciate the reviewer’s suggestion and have re-quantified our images. We have now included additional images of cells in non-mineralized and OIM-treated conditions in Supplementary Figure S15 to further support our statements. We have also modified the quantification of the dendrites to only include projections that were more than 5 µm in length and 1 µm in width, as to exclude filopodia and concentrate on longer dendrites.

Supplementary Figure S15: Quantification of the number (a) and length (b) of dendrite-like protrusions in hMSCs after 7 days of culture in 3D matrices. The cells were fixed, stained for F-actin and imaged using confocal microscopy (63x objective, Zeiss 210 Airyscan LSM 880). The mean number of protrusions per cell was counted manually using the cell counter plugin in imageJ software. The average protrusion length was assessed by manually tracing the extensions with freehand selection tool and their lengths were measured with ImageJ. Thin filopodial structures of length and width less than 5 µm and 1 µm, respectively were excluded from the data analysis for dendrite-like protrusions. Cells in the mineralized group exhibited a significant increase in the number and length of protrusions compared to non-mineralized and OIM groups. Data represented as Mean±SD, from four independent experiments with at least 30 cells per group. Comparison by one-way Anova (**p<0.01, ***p<0.001). Representative confocal images of cells from 4 different biological replicates exhibiting dendrite-like protrusions in 3D non-mineralized (c), mineralized (d) and OIM (e) constructs after 7 days. Small extensions stemming from the protrusions can be particularly noted in cells encapsulated within the mineralized matrix which may play a role in orienting dendrite formation and growth⁶⁰.

4. The reviewer does not agree that DMP1-expressing cells are more faint in non-mineralized hydrogels, nor that they are near the unstained vasculature capillaries. The differences in staining are very minimal between the two images, and many of the stained cells are far from the vascular-like channel. Some rephrasing here is warranted to temper the claims for this aspect of the study. We acknowledge the reviewer’s concern. We have modified the caption for Supplementary Figure S22 to read: **Supplementary Figure S22:** a) Engraftment of non-mineralized and (b) mineralized hydrogels in-vivo. H&E image showing erythrocyte filled capillaries (yellow arrow head) within the constructs after 3 weeks of implantation in nude mice. Scale bar: 500 µm. c) Expression of DMP1 was visible in (c) non-mineralized and (d) mineralized constructs implanted with pre-formed vasculature after 7 days of culture in-vitro, followed by 7 days in-vivo. DMP1-expressing cells (yellow arrows), which are also visible near cross-sectioned luminal structures,

support the presence of osteogenically differentiated hMSCs adjacent to vascular capillaries. Scale bar: 100 μm (N=4).

Some additional comments that could enhance this manuscript are provided below:

1. Introduction: It would be appropriate to rephrase the sentence regarding bone targeting diseases by stating “We further validate this model system by showing that the engineered tissue can stimulate homing of engrafted prostate cancer cells in-vivo”
The sentence has been modified as suggested.

2. The authors provide in S15 five confocal image sets for actin, collagen and merged staining, should this caption have read “Representative confocal images of cells from 5 different biological replicates”, or are these five sample images from these 4 samples (i.e. 2 imaged regions from one replicate)?

We apologize for the oversight. The Supplementary Figure S15 has been modified to show one representative image from each biological replicate under non-mineralized, mineralized and OIM-treated conditions.

3. It is this reviewers opinion that many of the important evidence in this paper is now contained in the supplementary figures (for example data S8 and S11, S12). It would be nice to see this data combined with figures in the main body of the paper.

Following reviewer’s suggestion, we have included day 7 alizarin red staining data from Supplementary Figure S11 in Figure 3. The relevant mechanical data discussed in S8 is included in Figure 1, and experts of representative images of OCN stained samples that are shown in S12 are also included in Figure 3c, thought for a different time point.

Figure 3: Osteogenic lineage commitment, gene and protein expression in mineralized cell-laden collagen. (a) Gene expression of osteocalcin (OCN) was significantly higher ($**p<0.01$) in mineralized collagen than in OIM treated, both of which were much higher than non-mineralized samples ($****p<0.0001$), after 21 days. Expression

of osteocyte-related genes (DMP1 and PDPN) was also comparable to OIM after 21 days of culture in mineralized collagen, and significantly higher at earlier time points (DMP1, $**p<0.01$ after 7 days, and $*p<0.05$ after 14 days). (N=3) (b) A significant increase in the expression of BMP 2 and BMP 6 for cells in both mineralized collagen and in OIM relative to non-mineralized controls is consistent with enhanced bone-specific metabolic activity. A marked increase in the ratio of RANKL/OPG in mineralized samples, however, suggests the stronger potency for cell-mediated bone-remodeling via a paracrine signaling in cell-laden mineralized constructs than in the other groups. (N=4 for BMPs and N=5 for others) At a cell-surface level, the expression of OCN and PDPN on day 14 was very low in (c,f) non-mineralized controls, and significantly higher for both (d,g) mineralized and (e,h) OIM-treated samples ($**p<0.01$, $***p<0.001$) (N=3). Similarly, surface expression of DMP1 was comparable for (k) OIM-treated cells ($***p<0.0001$) and (j) mineralized collagen ($*p<0.05$), both of which were significantly higher than (i) non-mineralized controls (N=4). Scale bar: 200 μm . Alizarin red staining to assay mineralization shows homogeneous and intense red staining even after 7 days in (m) mineralized constructs, while staining was more diffuse in (n) OIM-treated samples and almost non-existent in (l) non-mineralized controls. Scale bar: 100 μm Reflectance confocal microscopy images of F-Actin/DAPI stained hMSCs in both (o) non-mineralized, (p) mineralized, and (q) OIM treated cell-laden collagen, illustrate the dendritic-like extensions of cells after matrix mineralization, reminiscent of osteocyte-like morphology. The above images, with cells digitally removed (r-t), show formation of well-defined lacunae-like regions in the locations where cells resided in mineralized matrix (s). Scale bar: 30 μm . Quantification of dendrite-like projections is showing in Supplementary Figure S15. All comparisons used ANOVA/Tukey. Source data are provided as a Source Data file.

4. For the images provided in Figure 4(e-f) it is requested to clarify that these are for mineralised conditions. Are such images available for the non-mineralised and OIM conditions and, if not, could this figure move to the Supplementary material to make way for the interesting data that is now in S8, S11 and S12?
We appreciate the reviewer's interest for the supplementary data that we provided. However, we believe that it is important to highlight the novelty and relevance of the results shown in figure 4(e-f). Figure 4 shows the distinct morphology of cells embedded in fully mineralized collagen from a 3D perspective, and includes both structural and chemical information, since the images are collected in backscattered mode that illustrate mineral density. This information is not available in any other figure of the paper. Therefore, we strongly believe that the serial imaging of these samples under backscattered SEM warrants the merit of being placed in the main manuscript. Also, it is important to highlight that this is the first time that this type of 3D reconstruction of serial backscattered SEM images is reported in the literature, which again warrants its space in the main paper. The novelty and difficulty in collecting these images also explain their high cost and the reason why we only collected 2D slice images and control groups and did not create 3D reconstructions, which is a highly labor intensive and expensive process. In light of these observations, we strongly believe that figure 4 should be kept in the main paper and reserved the choice of maintaining it as it is.
5. S17: The figure heading reads "In (l) multiple narrow processes are shown in a cell (yellow arrow), and after 90 min, these processes have either disappeared or moved to another location (green arrows)". Presumably the intent here was to refer to (M) but this has not been indicated in the text of the heading.
We thank the reviewer for pointing this out and the oversight has been corrected.

REVIEWERS' COMMENTS:

Reviewer #2 (Remarks to the Author):

The authors have made final revisions to the manuscript, most notably by including a revised criteria for including cell projects to quantify dendrite formation and providing new data on this basis. The authors have included certain data from the supplementary figures (S8 and S11, S12) with figures in the main body of the paper. The authors have also revised the terminology to distinguish appropriately between dendrites and filopodia and addressed the other remaining minor comments. The paper now represents a comprehensive and extensive body of work that will influence thinking in the field.